



# A global view on atmospheric concentrations of sub-3 nm particles measured with the Particle Size Magnifier

Jenni Kontkanen[1], Katrianne Lehtipalo[1,2], Lauri Ahonen[1], Juha Kangasluoma[1], Hanna E. Manninen[1], Jani Hakala[1], Clémence Rose[3], Karine Sellegri[3], Shan Xiao[4], Lin Wang[4], Ximeng Qi[5], Wei Nie[5], Aijun Ding[5],
Huan Yu[6], Shanhu Lee[7], Veli-Matti Kerminen[1], Tuukka Petäjä[1], and Markku Kulmala[1]

[1]Department of Physics, University of Helsinki, 00014 Helsinki, Finland
[2]Paul Scherrer Institute, 5232 Villigen PSI, Switzerland
[3]Laboratoire de Météorologie Physique, UMR6016, CNRS/UBP, 63178 Aubière, France
[4]Shanghai Key Laboratory of Atmospheric Particle Pollution and Prevention (LAP3), Department of Environmental Science
& Engineering, Fudan University, 200433 Shanghai, China
[5]Joint International Research Laboratory of Atmospheric and Earth System Sciences, School of Atmospheric Sciences, Nanjing University, 210023 Nanjing, China
[6]School of Environmental Science and Engineering, Nanjing University of Information Science and Technology, Nanjing, China
[7]Department of Atmospheric Science, University of Alabama in Huntsville, Huntsville, Alabama

*Correspondence to*: Jenni Kontkanen (jenni.kontkanen@helsinki.fi)

**Abstract.** The measurement of sub-3 nm aerosol particles is technically challenging. Therefore, there is a lack of knowledge about the concentrations of atmospheric sub-3 nm particles and their variation in different environments. In this study, the concentrations of ~1–3 nm particles measured with a Particle Size Magnifier (PSM) were investigated at nine sites around the
world. Sub-3 nm particle concentrations were highest at the sites with strong anthropogenic influence. In boreal forest measured particle concentrations were clearly higher in summer than in winter, suggesting the importance of biogenic precursor vapors in this environment. At all sites sub-3 nm particle concentrations had daytime maxima, which are likely linked to the photochemical production of precursor vapors and the emissions of precursor vapors or particles from different sources. When comparing ion concentrations to the total sub-3 nm particle concentrations, electrically neutral particles were
observed to dominate in polluted environments and in boreal forest during spring and summer. Generally, the concentrations of sub-3 nm particles seem to be determined by the availability of precursor vapors rather than the level of the sink caused by pre-existing aerosol particles. The results also indicate that the formation of the smallest particles and their subsequent growth to larger sizes are two separate processes, and therefore studying the concentration of sub-3 nm particles separately in different size ranges is essential.

## 1 Introduction

The majority of atmospheric aerosol particles, in terms of their number, are formed via gas-to-particle conversion (Spracklen et al., 2006; Yu et al., 2010), often referred to as new particle formation (NPF). NPF contributes significantly to the global cloud condensation nuclei (CCN) budget, and thus affects the climate (Merikanto et al., 2009; Wang and Penner, 2009; Kazil et al., 2010; Makkonen et al., 2012). According to current knowledge, NPF proceeds via the formation of molecular clusters
from atmospheric vapors and their subsequent growth to larger sizes (Kulmala and Kerminen. 2008; Zhang et al., 2012; Kulmala et al., 2013; Kulmala et al., 2014). However, the chemical and physical processes leading to cluster formation and growth are not well understood, except in controlled systems in the laboratory (e.g. Kirkby et al., 2011; Almeida et al., 2013; Schobesberger et al., 2013; Kirkby et al., 2016; Lehtipalo et al., 2016; Tröstl et al., 2016). The continuous existence of ion clusters in the atmosphere has been known for decades (see Hirsikko et al., 2011 and references therein), while knowledge
about the concentrations and dynamics of atmospheric neutral clusters is more limited. There is evidence supporting the existence of neutral sub-3 nm particles and their importance in NPF, for example, in boreal forest (Kulmala et al., 2007, 2013),





whereas some modelling studies claim that ion-mediated mechanisms dominate NPF also there (Yu and Turco, 2000, 2008). Part of this controversy has been due to the inability to directly detect neutral clusters and sub-3 nm particles.

In recent years, the number of studies about the concentrations of atmospheric sub-3 nm particles has been increasing (Lehtipalo et al., 2010; Jiang et a., 2011; Kulmala et al., 2013; Yu et al., 2014; Rose et al., 2015; Xiao et al., 2015; Yu et al.,
2016; Kontkanen et al., 2016). However, the existing literature mainly comprises of concentration data from campaign measurements in specific environments. Also, the chemical composition and sources of sub-3 nm particles (natural or anthropogenic) and their precursors in different environments are still largely unknown. In addition, it is unclear if neutral sub-3 nm particles exist in all environments, or if ions dominate the sub-3 nm particle population in some conditions. Reflecting this, the terminology for sub-3 nm particles has also been variable: they have been called e.g. nano-CN, nano-particles, clusters
or seeds, depending on the reference (McMurry et al., 2011). For convenience, in this article we refer to all measured sub-3 nm particles as particles, even though some of them can be ions, clusters or even big molecules that are activated by the supersaturated vapor in condensation particle counters (CPCs), similarly as they could be activated in the atmosphere.

Until recently, measurements of atmospheric aerosol particles have been limited to the sizes above ~3 nm, which is the lowest detection limit of conventional ultrafine CPCs (McMurry, 2000). Sporadic measurements at smaller sizes have been reported,
often with custom-made or modified instruments (e.g. Mordas et al., 2008; Sipilä et al., 2008, 2009; Lehtipalo et al., 2009, 2010, 2011), but the data sets are not directly comparable due to differences in techniques and measured size ranges. The development of the Neutral cluster and Air Ion Spectrometer (NAIS) opened up the possibility to conduct systematic studies of ion concentrations down to 0.8 nm in mobility diameter, and of the total concentrations, including both charged and neutral particles, down to about 2 nm (Kulmala et al., 2007; Manninen et al., 2009; Mirme and Mirme, 2013). The lowest size limit
of the total concentration measurement with the NAIS is limited by the size distribution of the corona ions used to charge the sample (Manninen et al., 2011). The development of CPCs using diethylene glycol (DEG) as a condensing vapor, which started after the study by Iida et al. (2009), has pushed the cut-off size of CPCs down to about 1 nm (Vanhanen et al., 2011; Jiang et al., 2011; Kuang et al., 2012; Wimmer et al., 2013). Technically, all these CPCs consist of two stages: in the first stage DEG is used to pre-grow particles, and in the second stage particles are further grown and counted by a conventional CPC. This new
technology has played a key role in filling the gap between aerosol and mass spectrometric measurements and increasing the understanding of NPF starting from the molecular level (e.g. Kirkby et al., 2011; Kulmala et al., 2012; Almeida et al., 2013; Kulmala et al., 2013; Schobesberger et al., 2013; Kulmala et al., 2014; Kirkby et al., 2016). Here we focus on the measurements performed with the Particle Size Magnifier (PSM), which is the first commercially available DEG-based CPC (Vanhanen et al., 2011).

In this article, we review atmospheric measurements of sub-3 nm particle concentrations conducted with the PSM by different research groups. The measurement sites cover a wide range of different environments from a relatively clean boreal forest in Finland to very polluted Chinese megacities. Most of the data sets were obtained from 1–2 month intensive measurement campaigns, but we also present two longer-period and previously unpublished data sets from a boreal forest site and an urban environment in the southern Finland. These measurements allow us to investigate the seasonal variation of sub-3 nm particles.

The objective of this study is to provide the first global view on the concentrations and the dynamics of sub-3 nm particles. More specifically, we aim to get insights into i) the concentrations of sub-3 nm particles in different environments and their variation on a daily and seasonal basis, ii) the fraction of ions of all sub-3 nm particles, and iii) the possible sources and sinks of sub-3 nm particles.





## 2 Methods

### 2.1 The Particle Size Magnifier (PSM)

The PSM was developed at the University of Helsinki and later commercialized by Airmodus Ltd. The prototype instrument (PSMproto) and the measurement principle were introduced by Vanhanen et al. (2011). The first generation of the commercial

instrument is called A09, while the second generation with developments in the flow control and in the measurement software is called A10. The PSM A10, together with the Airmodus CPC A20 and the controlling software, is called the nano Condensation Nucleus Counter (nCNC) system (A11). For simplicity, we refer here to all instruments as PSMs, regardless of small differences in the design and the variability in the counter CPCs deployed at different measurement sites.

The operation principle of the PSM is based on a mixing-type CPC. The heated saturated flow is mixed turbulently with the

colder sample flow to create supersaturation in the mixing region. DEG vapor starts to condense on particles in the mixed flow, and the particles grow in the growth tube of the instrument until they reach diameters of about 90 nm. After that the particles are sampled into a regular CPC for the further growth by the condensation of another vapor (usually butanol), and they are finally counted by an optical detector. The advantage of a mixing-type design is that the mixing ratio of the saturated and sample flow can be quickly and accurately adjusted by changing the saturator flow rate. This affects the supersaturation

level which the particles encounter in the instrument, and thus the cut-off size of the instrument. Comparing concentrations measured with CPCs at different cut-off sizes has often been used as a method to estimate the concentration of particles in the size range between the cut-off sizes (Alam et al., 2003; Kulmala et al., 2007). The PSM can be operated in a so-called scanning mode, in which the saturator flow and therefore also the cut-off size is changed continuously, which enables the measurement of particle size distributions (Gamero-Gastãno and Fernández de la Mora, 2000; Vanhanen et al., 2011; Lehtipalo et al., 2014).

However, accurate calibrations are required for getting size information from measurements in the scanning mode.

Calibration methods in the sub-3 nm size range have evolved in recent years (Ude and Fernàndez de la Mora, 2005; Sipilä et al., 2009; Kangasluoma et al., 2014, 2015; Wimmer et al., 2015). Still, most of the calibrations are performed using electrically charged particles, as the size selection of particles is done based on their electrical mobility and an electrometer is used as the reference instrument for concentration. Winkler et al. (2008) showed that neutral particles need higher supersaturation to be

activated in a CPC than charged particles. In the PSM the difference in the cut-off diameter between neutral and charged particles is typically on the order of 0.5 nm (Kangasluoma et al., 2016a). Similarly, the chemical composition of particles affects their activation in the PSM. For inorganic particles the changes in the composition lead to about +/- 0.2 nm uncertainty in the cut-off size (Kulmala et al., 2013), while organic samples are activated clearly less efficiently by DEG vapor (Kangasluoma et al., 2014; Kangasluoma et al., 2016a).

The PSMs used in this study have been calibrated either with tetra-alkyl ammonium halides used as mobility standards (Ude and Fernàndez de la Mora, 2005), tungsten oxide particles (Vanhanen et al., 2011), or ammonium sulfate clusters (Wimmer et al., 2013; Kangasluoma et al., 2014). The operation temperatures of each instrument were adjusted during the initial calibration so that only a few counts from homogenous nucleation were allowed at the highest saturator flow rates. The PSMs used in Helsinki and in Hyytiälä in 2014–2016 had an automatic background measurement system (Kangasluoma et al., 2016b) and

they were thus allowed to have a higher background to maximize the activation efficiency for organic clusters. The background counts were subtracted from the data during the data analysis. The concentration range measurable with the PSM is mainly dependent on the counter CPC.

The data measured in the scanning mode of the PSM need to be inverted to get a size distribution. Two methods have been presented by Lehtipalo et al. (2014). The first one assumes a step-function like cut-off curve for each saturator flow rate of the

PSM. The difference in concentration between two flow rates (which determine the size bin limits) is corrected only by the





detection efficiency at the mean size of the bin to obtain the concentration in that size bin. This method resembles estimating the concentration of small particles by comparing the readings of two CPCs at different cut-off sizes. The second method takes into account the measured activation curves for each saturator flow rate and uses a non-negative matrix inversion routine to calculate the size distribution. This method tends to lead to slightly higher concentrations than the first method, which is partly

due to rejecting negative values resulting from fluctuations in the total concentration before data inversion and partly due to more accurate corrections for the detection efficiency, which can have a large effect at the smallest sizes. In addition, all the data sets have been corrected for particle losses in the sampling lines (Kulkarni et al., 2001).

The different PSMs used in this study had slightly different lowest and highest cut-off sizes, and different size bins were used during the data inversion due to individual settings and calibrations for each instrument. Due to this and the uncertainties in

determining the exact size limits, we chose to use the maximum size range available from the PSMs, which was from ~1 nm to ~2–3 nm for the PSMs operated in the scanning mode. If the largest size was smaller than 3 nm, or if the PSM had not been operated in the scanning mode, we obtained the ~1–3 nm concentration from the difference in the concentration measured with the PSM and another aerosol instrument with the cut-off size of 3 nm, i.e. a DMPS (Differential Mobility Analyzer) or a SMPS (Scanning Mobility Spectrometer). Although this way we aimed to obtain sub-3 nm particle concentrations for different sites

in as similar size ranges as possible, the reader should keep in mind that especially the differences in the lowest cut-off size of the PSMs can affect the comparability of data sets. Finally, when comparing different data sets, we calculated 30 min medians of all data. The time resolution of the PSM is 4 min in the scanning mode when averaging over an upward and a downward scan. The time resolution of the DMPS is typically 10 min, while the SMPS has a time resolution of about 3 min.

### 2.2 Measurement sites and instrumentation

In this study, PSM measurements from nine sites around the world were analyzed (Fig. 1). Measurements at each site are described below. In addition, the instruments used at different sites, the measurement time periods and the size ranges for particle measurements are summarized in Table 1.

### 2.2.1 Hyytiälä (HTL)

In Hyytiälä measurements were conducted at the SMEAR II station (Station for Measuring Forest Ecosystem-Atmosphere

Relations) in southern Finland (61° 5' N, 24° 17' E; 181 m above sea level) (Hari and Kulmala, 2005). The station is located about 200 km north of Helsinki. The closest urban area is the city of Tampere, which is located about 50 km southwest of the station and has the population of about 200 000. The station is surrounded by a Scots pine (*Pinus sylvestris*) forest, and monoterpenes dominate the emissions of biogenic volatile organic compounds (BVOCs) at the site (Rantala et al., 2015). The PSM measurements were conducted between 2010 and 2016. In the years before 2015 the measurements were shorter intensive

campaigns, while in 2015–2016 the measurement period covered one year (see Table 1 for the exact measurement periods). During the first measurement campaign in 2010 the prototype PSM was used, after that in 2011 and 2012 the PSM model A09, in 2013 the PSM A10 and in 2014 and 2015 the PSM A11 (which is similar to A10). The size bins used in the inversion were slightly different for different measurement campaigns as different instruments were used: the lowest cut-off size varied between 1.0 and 1.3 nm and the highest cut-off size between 2.0 and 2.5 nm (see Table 1). In 2010–2013 the sampling of

particles was done with a 40 cm long inlet tube (6 mm in diameter) with 2.5 lpm (liters per minute) flow rate. Starting from 2014, the inlet specially designed to minimize sampling losses was used (Kangasluoma et al., 2016b). In this inlet, a flow of 7.5 lpm was taken directly from outside air through a 40 cm long tube and the actual sample (2.5 lpm) was taken from the middle of the flow with a probe (core sampling). An automatic background measurement was performed every third hour using filtered ambient air.





In addition to the PSM, measurements with a twin-DMPS system (Aalto et al., 2001) were utilized. The DMPS system measured the particle size distribution between 3 and 1000 nm. Thus, by subtracting the total particle concentration measured with the DMPS from the concentration measured with the highest cut-off size of the PSM, the particle concentration in the size range of ~2–3 nm was obtained. Furthermore, the ion size distribution between 0.8 and 42 nm was measured with the NAIS (Manninen et al., 2016). From these measurements, the concentration of sub-3 nm ions was obtained. The results of PSM measurements conducted during spring 2011 in Hyytiälä have been published by Kulmala et al. (2013).

### 2.2.2 Helsinki (HEL)

In Helsinki, measurements were performed at the SMEAR III station (60º 12' N, 24º 58' E; 26 m above sea level) (Järvi et al., 2009). The city of Helsinki is located on the southern coast of Finland. The Helsinki Metropolitan area, consisting of Helsinki and the neighboring municipalities, has the population of about 1.4 million. The measurement station is situated on a hill next to the university campus, about 5 km north of the Helsinki city center. The surroundings of the station are heterogeneous, including buildings, parking lots, roads, deciduous forests and low vegetation (for a more detailed description, see Järvi et al. (2009)). The measurements with the PSM A11 took place in 2015, covering one year (see Table 1). For the data inversion, the size bins of 1.1–1.3 nm, 1.3–1.5 nm, 1.5–2 nm were used, which were identical with the PSM operated in Hyytiälä in 2015–2016. The sampling of the PSM was done by using an inlet system identical to the inlet used in Hyytiälä after 2014, including a core sampling probe and automatic background measurements (Kangasluoma et al., 2016b). Similarly to in Hyytiälä, a twin-DMPS system (Aalto et al., 2001) was used to measure the particle size distribution in the size range from 3 to 800 nm. Therefore, the concentration between 2 and 3 nm was obtained by subtracting the total particle concentration measured with the DMPS from PSM measurements.

### 2.2.3 San Pietro Capofiume (SPC)

The San Pietro Capofiume meteorological station is situated in northern Italy (44º 39' N, 11º 37' E; 11 m above sea level). The station is located in the Po Valley with high emissions of anthropogenic pollutants, about 30 km northeast of the city of Bologna. The surroundings of the site are flat and homogeneous, mainly consisting of harvested fields (Decesari et al., 2001). The measurements with the PSM A09 were conducted at the station in June–July 2012 during the PEGASOS (Pan-European Gas–Aerosol–Climate Interaction Study) Zeppelin campaign (see Table 1). The cut-off sizes of the PSM were 1.5 and 1.8 nm. In addition to the PSM, a twin-DMPS system covering the size range of 3–600 nm (Laaksonen et al., 2005) was operated. Thus, by combining PSM and DMPS measurements the particle concentration in the size range of 1.5–3 nm was obtained. In addition, the ion concentration for the same size range was obtained from NAIS measurements. The results of these measurements, focusing on the analysis of NPF events, have been presented by Kontkanen et al. (2016).

### 2.2.4 Puy de Dôme (PDD)

The Puy de Dôme measurements site is located at the top of the Puy de Dôme mountain in central France (45º 46' N, 2º 46' E, 1465 m above sea level). The station is surrounded by fields and forests. The closest town, Clermont-Ferrand, is located about 16 km east of the mountain at 396 m above sea level and has the population of about 150 000. See Venzac et al. (2009) for a more detailed description of the measurement site. The PSM A09 was operated at the station in January–February 2012 (see Table 1). From the PSM measurements, the particle concentration between 1.0 and 2.5 nm was obtained. Furthermore, the ion concentration in the same size range was obtained from NAIS measurements. An SMPS measuring the particle size distribution between 10 and 420 nm was also operated at the site. The data measured when relative humidity (RH) exceeded 98% was omitted from the analysis, as high values of RH indicate that the station was inside a cloud (Rose et al., 2015). Rose





et al. (2015) have published the results of this measurement campaign, concentrating on NPF events in the free troposphere (FT) and at the interface between the boundary layer and FT.

### 2.2.5 Kent (KNT)

The measurements in Kent, Ohio, were conducted at the Kent State University's campus (41º 9' N, 81º 22' W, 320 m above sea level). Kent is a small Midwestern town with about 30 000 inhabitants. The closest larger cities are Akron (30 km west of Kent), Cleveland (60 km north-west), and Pittsburgh (100 km east). Measurements with the PSM A09 were performed between December 2011 and January 2012 (see Table 1). The PSM was operated with the fixed saturator flow, corresponding to the cut-off size of ~1 nm. During the measurements, the ambient air was drawn at a flow rate of 3000 lpm into an air duct (1.5 m long and 10 cm in diameter) to which the PSM was directly connected via a 4 cm tube (0.64 cm in diameter). In parallel to

the PSM measurements, the concentrations of particles in the size range of 3–478 nm was measured with the combination of two SMPSs (see Yu et al. (2014) for details). Thus, the concentration of particles between 1 and 3 nm was obtained by subtracting the total particle concentrations measured with the SMPSs from concentrations measured with the PSM. The results of these measurements, together with the measurements from Brookhaven (see below), have been presented in Yu et al. (2014).

### 2.2.6 Brookhaven (BRH)

In Brookhaven, New York, measurements were performed at Brookhaven National Laboratory in Long Island (40º 52' N, 72º 53' W, 24 m above sea level). The site is located 80 km east of New York City. The Long Island Sound is 16 km north of the site and the coast of Atlantic Ocean 25 km south. The measurement site is located within an urban neighborhood and surrounded by a mixed deciduous forest. The measurements with the PSM A09 were conducted in July–August 2011 (see Table 1). Similarly to in Kent, the PSM had a fixed cut-off of about 1 nm, and the particle concentration between 1 and 3 nm

was obtained by combining PSM measurements with the SMPS measurements. During the measurements, ambient air was drawn into the instrument container at a flow rate of 150 lpm using a 2.1 m tube (5.08 cm in diameter) and the PSM sampled from a split flow of 30 lpm with a 30 cm tube (2.54 cm in diameter). See Yu et al. (2014) for the more detailed description of these measurements.

### 2.2.7 Centreville (CTR)

The Centreville measurement site is located in Brent, Alabama (32° 54' N, 87° 15' W, 139 m above sea level). The landscape surrounding the research site is a mix of agricultural lands and mixed deciduous forests. Isoprene is the dominant BVOC emitted from forests, while agricultural lands have low emission rates of isoprene and monoterpenes. The strongest pollutant emission sources of $NO_x$ (17 000 tons $yr^{-1}$) and $SO_2$ (92 000 tons $yr^{-1}$) in the state of Alabama are located within 100 km of the site. The measurements with the PSM A09 were conducted at the site during the SOAS (Southern Oxidant and Aerosol Study)

campaign in June–July 2013 (see Table 1). From the PSM measurements the particle size distribution between 1.1 and 2.1 nm was obtained. In addition, two SMPSs (TSI 3936) were used to measure the particle size distribution in the combined size range from 3 to 740 nm. One SMPS had a Nano Differential Mobility Analyzer (Nano-DMA, TSI 3085) and a TSI 3786 water Condensation Particle Counter (CPC). The second SMPS had a long-DMA (TSI 3081) coupled with a TSI 3772 butanol CPC.

### 2.2.8 Shanghai (SH)

The measurements in Shanghai were conducted on the campus of Fudan University (31º 18' N, 121º 30' E) at about 20 m height from the ground. The site is located north-east of the center of Shanghai, which is the largest city in China with about 24 million inhabitants. One of the city's main highways is located 100 m south of the measurement site. The measurements with the PSM A11 were performed between November 2013 and January 2014 (see Table 1). The PSM was operated in the





scanning mode, and the particle concentration between 1 and 3 nm was obtained from the measurements. In addition, two SMPSs were used to measure the particle size distribution between 3 and 615 nm. During the measurements, ambient air was drawn at a flow rate of 4332 lpm into a 5.0 m manifold (10.16 cm in diameter). From this manifold, air was drawn at a flow rate of 1.75 lpm through a 18 cm tube (0.64 cm in diameter) and diluted with zero air flow at a ratio of 1:1 before entering the

PSM. Xiao et al. (2015) have published the results of these measurements, discussing especially the formation and growth rates of particles.

**2.2.9 Nanjing (NJ)**

In Nanjing measurements were performed at the Station for Observing Regional Processes of the Earth System (SORPES), which is situated about 20 km east of suburban Nanjing (Ding et al., 2013). The site is located on top of a hill on the Xianlin

campus of Nanjing University (118° 57'E, 32° 07' N; 40 m above sea level). The measurements with the PSM A11 were conducted between December 2014 and January 2015 (see Table 1). The PSM was operated in the scanning mode and five size bins between 1 and 3 nm were used for the inversion. In addition, AIS (Air Ion Spectrometer; Mirme et al., 2007) measurements were conducted, providing ion concentrations in the same size range (Hermann et al., 2013). The particle size distribution between 6 and 800 nm was also measured with a DMPS (Qi et al., 2015).

**2.2.10 Supporting data**

In addition to the measurements of sub-3 nm particle and ion concentrations, other data recorded at the measurement sites were utilized in the analysis. These data included different meteorological variables (e.g. temperature, RH, and radiation) and trace gas concentrations (e.g. $SO_2$ and $NO_x$). Condensation sink (CS), which describes the loss rate of vapor due to condensation on pre-existing aerosol particles (Kulmala et al., 2001), was calculated from particle size distribution data measured with the

DMPS or the SMPS. In addition, the concentration of sulfuric acid was measured with a CIMS (Chemical Ionization Mass Spectrometer; Eisele and Tanner, 1993) in Kent and Brookhaven (Yu et al., 2014), and in Hyytiälä during spring 2011 (Kulmala et al., 2013). For other measurement campaigns, sulfuric acid concentration was estimated using a proxy. For Hyytiälä the proxy in Petäjä et al. (2009) was used, as it has been derived and validated with measurements from this specific site. For other measurement sites, the proxy presented in Mikkonen et al. (2011) was utilized as it has been developed based on data from

several different measurements sites.

**3 Results and discussion**

**3.1 Sub-3 nm particle concentrations and their variation at different sites**

**3.1.1 Sub-3 nm particle concentrations at different sites**

The concentration of sub-3 nm particles was observed to vary significantly at each measurement site and between different

environments. The medians (and different percentile ranges) of sub-3 nm particle concentration at different measurements sites are shown in Fig. 2 (see also Table 2). The concentration was highest at the sites with strong anthropogenic influence in Nanjing and Shanghai, China, and in San Pietro Capofiume, Italy. The median sub-3 nm particle concentration was $1.7 \times 10^4$ $cm^{-3}$ in Nanjing, and $8.5 \times 10^3$ $cm^{-3}$ in Shanghai and San Pietro Capofiume. High concentrations were also observed at the urban site in Helsinki, Finland, where the median concentration was $5.8 \times 10^3$ $cm^{-3}$. At the Finnish boreal forest site, Hyytiälä, the

median concentration, calculated from all the data measured in 2010–2016, was lower than in Helsinki, $2.0 \times 10^3$ $cm^{-3}$. The lowest sub-3 nm particle concentrations were observed at the French mountain site, Puy de Dôme, with the median





concentration of $5.0 \times 10^2$, and at North American sites, Kent, Brookhaven and Centreville, where the median concentrations were $4.7 \times 10^2$ cm$^{-3}$, $8.0 \times 10^2$ cm$^{-3}$, and $5.9 \times 10^2$ cm$^{-3}$, respectively.

The observed differences in sub-3 nm particle concentrations indicate that their formation is generally favored in polluted environments (see Sect. 3.4.1 where sulfuric acid concentration and condensation sink at different sites are compared). This
can be explained by the high concentrations of low-volatile precursor vapors, which originate from e.g. fuel combustion and traffic, and can form small particles in the atmosphere (e.g. Arnold et al., 2012; Karjalainen et al., 2015; Sarnela et al., 2015). Some of the traffic-related particles may also be primary and formed inside vehicle engines (Jayaratne et al., 2010; Karjalainen et al., 2014; Alanen et al., 2015). At sites with lower anthropogenic influence, like Puy de Dôme, lower sub-3 nm particle concentrations were observed, which is likely due to the lower concentrations of precursor vapors and the absence of primary
particle sources. On the other hand, in pristine environments the emissions of organic vapors from vegetation may promote the formation of sub-3 nm particles (Ehn et al., 2014). Interestingly, sub-3 nm particle concentration was clearly higher in a Finnish boreal forest, where BVOC emissions are dominated by monoterpenes, than in Centreville, the southeastern US, where isoprene emissions dominate (Xu et al., 2015). Earlier, Kanawade et al. (2011) observed that NPF events are less frequent in mixed deciduous forests than in boreal forests, which they attributed to high emissions of isoprene.

When comparing concentrations between different sites, the median particle concentrations observed in Brookhaven and Kent can be considered relatively low compared to other urban sites. The low concentrations may indicate that the particle activation efficiency inside the PSM was during these measurements lower than in other measurement campaigns, which can be due to, for example, particle composition or technical reasons. Furthermore, one should note that measurements at different sites were conducted at different times of the year. Therefore, a possible seasonal variation in sub-3 nm particle concentration due to the
variation in their sources and sinks may bias the comparison. For example, in Kent and Puy de Dôme, where the median concentrations were lowest of all sites, the measurements were conducted in winter when the photochemical production of precursor vapors and the emissions of biogenic vapors can be expected to be lower than in summer. The boundary layer dynamics may also affect concentrations especially at the high-altitude Puy de Dôme site: in winter the station is often above the boundary layer, which prevents the transport of precursor vapors from near-ground sources to the site (Venzac et al., 2009).

**3.1.2 Interannual variability in sub-3 nm particle concentration in Hyytiälä**

Figure 3 presents sub-3 nm particle concentrations in Hyytiälä during different measurement campaigns starting from the first field measurements performed with the PSM in 2010 (see also Table 3). The data from 2015–2016, covering one year, were divided into spring (March–May), summer (June–August), autumn (September–November) and winter (December–February) to enable the comparison to other years' shorter measurement periods. Sub-3 nm particle concentration seems to have a clear
seasonal variation in Hyytiälä (see also Sect. 3.1.3 and 3.2.2). The median concentrations were higher during measurements performed in spring and summer ($9.4 \times 10^2$–$5.4 \times 10^3$ cm$^{-3}$) than in autumn and winter ($5.8 \times 10^2$–$1 \times 10^3$ cm$^{-3}$). The measurements from different years agree rather well, despite the differences in the instrument model and the exact settings of the PSM and the sampling lines, which can affect the cut-off size of the instrument and particle losses. It needs to be noted, though, that in spring 2016 sub-3 nm particle concentration was on average lower than in other springs. This may be related to untypical
environmental conditions, as the frequency of NPF events was clearly lower in that spring compared to other years (Table 3). The connection between sub-3 nm particle concentrations and environmental conditions is further discussed in Sect 3.4. Generally, the median value and the variation of sub-3 nm particle concentration observed in the spring campaigns compare well to the concentrations reported by Lehtipalo et al. (2009, 2010), who measured 1.5–3 nm particles with a pulse-height CPC in Hyytiälä during spring 2007 and 2008.



### 3.1.3 Particle concentrations in different size bins in Hyytiälä and Helsinki

In addition to the total sub-3 nm particle concentration, the concentrations in different sub-3 nm size bins were investigated for Helsinki and Hyytiälä. For this, the data sets from 2011 and 2015–2016 were used, as they had almost identical size bins: 1.1–1.3 nm, 1.3–1.5 nm, 1.5–2 nm, and 2–3 nm in 2015–2016 and 1.1–1.3 nm, 1.3–1.5 nm, 1.5–2.1 nm, and 2.1–3 nm in 2011.

To investigate the seasonal variation of particle concentrations, the data sets were divided into spring (March–May), summer (June–August), autumn (September–November) and winter (December–February). The particle concentrations in different size bins in these seasons are presented in Table 4. Note that, for clarity, only the size bin limits used in 2015–2016 are marked in the table.

A seasonal variation in particle concentrations was observed at both sites. In Hyytiälä, sub-3 nm particle concentration was

higher in summer and spring than in winter and autumn. In the sub-2 nm size bins the concentrations were highest in summer; this was clear especially in the smallest size bin (1.1–1.3 nm) where the median concentration was 1129 cm$^{-3}$ in summer and 235 cm$^{-3}$ in winter. In the largest size bin (2–3 nm) the highest concentrations were detected in spring, with the median concentrations ranging from 109 to 300 cm$^{-3}$ in different seasons. The summer-time maximum in the concentration of the smallest particles is likely related to the strong photochemical production of precursor vapors and the high emissions of organic

vapors from vegetation at this time of the year. Seasonal differences were observed also in the ratio of 1.1–2 nm to 2–3 nm particle concentrations in Hyytiälä. In summer and autumn, the 1–2 nm particle concentration was 6–8 times higher than the concentration in the 2–3 nm size range, while in spring and winter the difference was only a factor of 2–2.5. This may indicate that in summer and autumn sub-2 nm particles are not able to grow to sizes larger than 2 nm efficiently. In spring, environmental conditions in Hyytiälä are favorable for particle growth, as indicated by frequent NPF events (Table 1; Dal

Maso et al., 2005), which probably explains why the difference between the size bins was then smaller. In winter, on the other hand, particle concentrations were low in all size bins.

In Helsinki differences in particle concentrations between different seasons were less distinct than in Hyytiälä. The highest concentrations were detected in spring and winter. In the smallest size bin (1.1–1.3 nm) the median concentration varied between 1134 and 2324 cm$^{-3}$ in different seasons and in the largest size bin (2–3 nm) between 1870 and 2214 cm$^{-3}$. The high

winter-time concentrations suggest that in Helsinki the formation of sub-3 nm particles is unlikely driven by the emissions of organic compounds from biogenic sources. In Helsinki the ratio of 1.1–2 nm to 2–3 nm particle concentrations varied between 1 and 2 in different seasons and it was highest in winter and spring and lowest in summer and autumn. The lower value of this ratio in Helsinki compared to Hyytiälä indicates that particle growth may be favored in an urban environment with stronger anthropogenic influence compared to clean boreal forest (Kulmala et al., 2005). It should be kept in mind, though, that the

composition of particles can be different in different environments, which can affect their activation probability in the PSM.

### 3.2 Diurnal variation of sub-3 nm particle concentration

### 3.2.1 Diurnal variation at different sites

The median diurnal variation of sub-3 nm particle concentration at different measurement sites is presented in Fig. 4. The differences in concentrations between different sites are obvious also here: sub-3 nm particle concentration was high at sites

with strong anthropogenic influence and lower in cleaner environments. Generally, sub-3 nm particle concentrations were highest during daytime and lowest at night. Still, at many sites moderate concentrations were observed also at night. In Nanjing, Shanghai, and San Pietro Capofiume, the daily minimum concentrations were between 6×10$^3$ and 8×10$^3$ cm$^{-3}$, and the daily maximum concentrations were between 2×10$^4$ and 5×10$^4$ cm$^{-3}$. At these sites particle concentrations had also weak secondary maxima in the evening. In Helsinki, sub-3 nm particle concentration typically varied between 2×10$^3$ and 1×10$^4$ cm$^{-3}$ during the





day. At this site the daytime maximum was wide, as the concentration was high from around 8:00 to 16:00 local time. In Hyytiälä the median diurnal cycle was not as strong as at other sites: particle concentration varied between $1.4 \times 10^3$ and $2.7 \times 10^3$ cm$^{-3}$. It needs to be noted, though, that in Hyytiälä the diurnal cycle had a strong seasonal variation, which is discussed in the next section (3.2.2). In Kent, Brookhaven and Puy de Dôme sub-3 nm particle concentrations were also lowest early in the

morning (~200–300 cm$^{-3}$), and highest around noon (~1–1.7 $\times 10^3$ cm$^{-3}$). In Brookhaven particle concentration additionally had a secondary maximum of about 900 cm$^{-3}$ in the evening after 19:00. In Centreville sub-3 nm particle concentration had a minimum early in the morning (about 300 cm$^{-3}$) and two separate maxima when the concentration reached about 900 cm$^{-3}$; the first maximum occurred before noon and the second in the evening around 20:00.

The observed daytime maxima in sub-3 nm particle concentrations likely result from the photochemical production of low-
volatile precursor vapors during daytime and the emissions of precursor vapors, and possibly also primary particles, from different anthropogenic and biogenic sources. Daytime maxima in sub-3 nm particle concentrations have also been reported in previous studies (Kulmala et al., 2013; Yu et al., 2014, 2016; Xiao et al., 2015; Rose et al., 2015; Kontkanen et al., 2016), where they have often been linked to NPF events. On the other hand, the daytime increase in sub-3 nm particle concentration is not necessarily followed by an NPF event where particles grow to large sizes (Yu et al., 2014, 2016; Xiao et al., 2015; see
Sect. 3.5). At urban sites, the diurnal cycle of sub-3 nm particle concentrations can be affected by variation in traffic conditions and other anthropogenic activities, which could explain the wide maximum in particle concentration observed in Helsinki. Furthermore, in Puy de Dôme the diurnal variation of sub-3 nm particle concentration can be influenced by the diurnal cycle of the boundary layer height, affecting the transport of precursor vapors to the site (Venzac et al., 2009; Rose et al., 2015).

The relatively high particle concentrations observed at many sites at night suggest that the formation of sub-3 nm particles
may also occur in the absence of solar radiation. This may imply the importance of low-volatile precursor vapors originating from the oxidation of, for example, organic compounds by ozone or nitrate radical (Ehn et al., 2014). Kirkby et al. (2016) observed in their chamber experiments that α-pinene ozonolysis products can form new particles efficiently even in the absence of sulfuric acid. In Hyytiälä high concentrations of sub-3 nm particles in the evening have been reported earlier (Lehtipalo et al., 2009), and they have been proposed to be related to the ozonolysis products of monoterpenes (Lehtipalo et al., 2011).
Evening maxima are frequently observed also in sub-3 nm ion concentrations in Hyytiälä (Junninen et al., 2008; Buenrostro Mazon et al., 2016). In Brookhaven Yu et al. (2014) found that the elevated concentration of sub-3 nm particles at night were linked to marine air masses and they were probably not connected to the oxidation of monoterpenes. In Centreville, where the evening maximum in sub-3 nm particle concentration was most distinct, BVOC emissions are dominated by isoprene (Xu et al., 2015). In earlier measurements in an isoprene-rich deciduous forest, the concentrations of 3–10 nm particles were observed
to increase in the evening when SO$_2$ concentration was high (Kanawade et al., 2011).

### 3.2.2 Diurnal variation in Hyytiälä and Helsinki in different seasons

To study the diurnal variation of sub-3 nm particle concentration in different seasons, the data sets from Hyytiälä and Helsinki (from where longer times series were available) were divided into spring, summer, autumn, and winter. Figure 5 illustrates the median diurnal cycle of sub-3 nm particle concentration in Hyytiälä and Helsinki in these seasons.

In Hyytiälä, the diurnal cycle of sub-3 nm particle concentration was stronger in spring and summer than in autumn and winter. In spring particle concentration started to increase from the nighttime value of about $2 \times 10^3$ cm$^{-3}$ after 5:00 and reached the maximum of about $5 \times 10^3$ cm$^{-3}$ around 14:00. In summer particle concentration did not increase as strongly during daytime: it varied between $1.4 \times 10^3$ cm$^{-3}$ observed at night and $2.8 \times 10^3$ cm$^{-3}$ detected around noon. In autumn and winter particle concentration stayed relatively low throughout the day: in autumn the concentration varied between $5 \times 10^2$ and $1 \times 10^3$ cm$^{-3}$, and

in winter between $4\times10^2$ and $7\times10^2\,\mathrm{cm^{-3}}$. The more pronounced daytime maximum in spring and summer than in other seasons is likely related to the stronger production of low-volatile precursor vapors in these months.

In Helsinki the daytime maximum in sub-3 nm particle concentration was distinct in all seasons. Sub-3 nm particle concentration started to rise after 5:00 from the nighttime level ($1.5$–$2.7\times10^3\,\mathrm{cm^{-3}}$ in different seasons) and was highest around

midday ($8.5\times10^3\,\mathrm{cm^{-3}}$–$1.8\times10^4\,\mathrm{cm^{-3}}$ in different seasons), and started to decrease again after 16:00. The highest concentrations were obtained in spring and lowest in autumn. The similarities in the diurnal cycle of sub-3 nm particle concentration in different seasons suggest that in Helsinki sub-3 nm particle concentrations are more affected by anthropogenic sources of precursor vapors, which typically are fairly constant throughout the year, than the emissions from biogenic sources. The fact that particle concentration started to rise in the morning at the same time in all seasons implies that the increase is not triggered

by photochemistry. Instead, it may be related to the morning traffic on the nearby roads. This hypothesis is supported by the results of Järvi et al. (2008), who found that the traffic rate on the road close to the SMEAR III station increased in the morning after 5:00, which coincided with the rise in black carbon concentration at the station.

### 3.2.2 Diurnal variation of particle concentration in different size bins

The median diurnal cycle of particle concentration in Helsinki and Hyytiälä was also studied in different sub-3 nm size bins

(Fig. 6). For this, only data from the years 2011 and 2015–2016 were utilized as the size bins were most comparable with each other in these years (see Sect. 3.1.3).

In Hyytiälä clear differences in the diurnal cycle of particle concentrations in different size bins were observed. In the size bin of 1.1–1.3 nm the particle concentration had a strongest diurnal cycle in summer: the concentration had a minimum in the early morning after which it increased and stayed high between 10:00 and 21:00. In other seasons, the 1.1–1.3 nm particle

concentration stayed more stable. In the size bin of 1.3–1.5 nm the particle concentration did not have a strong diurnal cycle in any season. However, in the size bins of 1.5–2 nm and 2–3 nm the particle concentration increased during daytime in spring. This increase is probably linked to NPF events, which are most frequent in Hyytiälä in spring (Table 2). On the other hand, the high daytime concentrations of 1.1–1.3 nm particles in summer likely result from the strong production of precursor vapors, originating from biogenic sources, at this time of the year. When comparing the diurnal cycles in different size bins to the

diurnal cycle of the total sub-3 nm particle concentration in Hyytiälä (Fig. 5), one can see that in summer the daytime maximum in the total sub-3 nm particle concentration was caused by the maximum in the concentration of the smallest, 1.1–1.3 nm, particles, while in spring the daytime peak was mostly due to the increase in the concentration of the largest, 2–3 nm, particles.

In Helsinki the diurnal cycles were quite similar in different size bins and during different seasons. In all size bins, particle concentrations were lowest in the early morning (around 4:00), after which they started to increase reaching the maximum

around midday and started to decrease again after 16:00. In spring the daytime peak values were higher than in other seasons; this was clear especially in the size bins of 1.1–1.3 nm and 1.5–2 nm. On the other hand, nighttime particle concentrations in sub-2 nm size bins were highest in winter. In the 2–3 nm size bin, the diurnal cycles in different seasons were almost identical.

### 3.3 Ratio of ions to total sub-3 nm particle concentrations

### 3.3.1 Ion ratio at different sites

To determine how large fraction of measured sub-3 nm particles were electrically charged, we studied the ion ratio, i.e. the ratio of sub-3 nm ion concentration measured with the NAIS to the total sub-3 nm particle concentration measured with the PSM in the corresponding size range. The ion ratios at different sites (only those with ion measurements available) are depicted in Fig. 7 (see also Table 2). Generally, the ion ratios were mainly determined by the total sub-3 nm particle concentration, as





the ratios were lowest at the sites with the highest total concentrations and highest at sites with the lowest total concentrations. This results from the smaller variation in ion concentrations between different environments than in the total particle concentrations: the median sub-3 nm ion concentrations (calculated for the same size ranges as the total particle concentration) were between 300 and 700 cm$^{-3}$ at different sites. The observed smaller variation in ion concentrations is consistent with earlier

observations by Manninen et al. (2010), and it can be explained by the fact that ion concentrations at the ground level are generally limited by ion production rates (Hirsikko et al., 2011). In San Pietro Capofiume, the median ion to the total particle concentration ratio was as low as 0.004 and in Nanjing 0.02. At other sites the ion ratios were higher. In Centreville the median ion ratio was 0.47, and in Puy de Dôme 0.60. On the other hand, Rose et al. (2015) showed that neutral particles dominate in Puy de Dôme during NPF events. In Hyytiälä, the ion ratio exhibited a strong seasonal variation. The median ion ratio was

rather low in spring and summer, 0.16 in spring and 0.33 in summer, which is consistent with the high total sub-3 nm particle concentrations observed in these seasons. In autumn the median ion ratio was 0.83 and in winter 0.71, and the ratio often exceeded unity in these seasons (see the discussion below). Lehtipalo et al. (2009, 2010) observed the ion ratio to be only about 0.01 in springtime in Hyytiälä by comparing ion concentrations measured with a BSMA to the total concentrations from a pulse-height CPC, but their measurements started only at about 1.3–1.5 nm.

The ion ratios exceeding unity, observed in Hyytiälä, Puy de Dôme and Centreville, are not physical, and thus indicate that the PSM is not able to detect all the 1–3 nm ions and particles. This may be caused by uncertainties in the detection efficiency due to the composition of particles and changing environmental conditions (see Sect. 2.1). For instance, the detection efficiency of the PSM has been observed to decrease when the relative humidity of the sample flow is decreased (Kangasluoma et al., 2013) and to be lower for organic clusters than for sulfate containing clusters (Kangasluoma et al., 2014, 2016a). As sub-3 nm

particles can originate from many different kinds of sources, their composition likely also varies, and therefore finding a representative calibration compound is difficult. Further work is still needed to consider these issues when conducting field measurements with the PSM. In addition, inaccuracies in ion concentrations measured with the NAIS may also cause uncertainties in ion ratios (Wagner et al., 2016).

Finally, it needs to be noted that the observed ion ratios depend strongly on the limits of the studied size range. This is due to

the pool of small ions which is constantly present in the atmosphere because of ionization of air molecules (e.g. Hirsikko et al., 2011). For example in Hyytiälä the median size of this ion pool is about 1.1–1.3 nm (Manninen et al., 2009). Thus, the observed differences in the ion ratio between different measurement sites and different measurement campaigns in Hyytiälä (see Tables 2 and 3) can partly be due to differences in the studied size ranges. In the next section the ion ratio in Hyytiälä is studied separately in different sub-3 nm size bins.

**3.3.2 Ion ratio in Hyytiälä in different size bins**

Table 4 shows the ratios of ion concentrations to the total particle concentrations separately in four sub-3 nm size bins in Hyytiälä (data only from the years 2011 and 2015–2016). The ratio was highest in the size bins below 2 nm, as anticipated due to the constant pool of small ions (Manninen et al., 2009; Hirsikko et al., 2011). In the smallest size bin, 1.1–1.3 nm, the ratio was lowest in summer (median value 0.34) when the total particle concentration in that size bin was high. In spring the median

ratio in this size bin was 0.75 and in autumn and winter 1.05. In the next two size bins (1.3–1.5 nm and 1.5–2 nm) the ion ratio was high in all seasons, with the median ratios ranging between 0.8 and 1.5. This further demonstrates that the PSM does not detect all sub-2 nm particles, as discussed in the previous section (3.3.1). On the other hand, it should be noted that during spring 2016 sub-3 nm particle concentrations observed in Hyytiälä were lower than in other years (see Table 3). This can partly explain the rather high value obtained for the springtime ion ratio. In the largest size bin, 2–3 nm, the ion ratio was low in all

seasons, with the median ratios varying between 0.03 and 0.07. This is expected, as at this size range most ions originate from





diffusion charging of neutral aerosol particles by collisions with the small ions, or from ion-induced nucleation. In Hyytiälä 2–3 nm ions have been observed to exist almost only during NPF events (Leino et al., 2016).

### 3.3.3 Diurnal variation of the ion ratio

The median diurnal cycles of the ratio of ion concentration to the total sub-3 nm particle concentration at different sites are illustrated in Fig. 8. At all sites the ion ratio was lowest during daytime and highest early in the morning, having the opposite diurnal cycle to that of the total sub-3 nm particle concentration (see Fig. 4). In San Pietro Capofiume the ratio varied between 0.003 and 0.008 (median values), reaching the highest value in the morning around 6:00. Kontkanen et al. (2016) also observed the morning maximum in the ion ratio in San Pietro Capofiume and proposed that it was caused by the earlier increase in ion concentration than in the concentration of neutral particles during NPF events. In Nanjing the ion ratio had a minimum around midday (0.007) and a maximum in the morning (0.05). Similarly, in Puy de Dôme the ion ratio was lowest, 0.14, around midday and highest, 1.3, in the morning. In Centreville the ion ratio varied between 0.28 and 0.95, being highest at night and lowest during daytime. As discussed in the previous section, in Hyytiälä the ion ratio was lowest in spring and summer and highest in autumn and winter. In spring and summer the ion ratio was lowest around noon and highest in the early morning, varying between 0.09 and 0.24 in spring and between 0.23 and 0.43 in summer. In autumn and winter the diurnal variation of the ratio was weaker: in autumn the ratio varied between 0.68 and 0.92 and in winter between 0.56 and 0.91.

Figure 9 presents the median diurnal cycles of the ratio of the ion concentration to the total particle concentration in different sub-3 nm size bins in Hyytiälä in different seasons. In the size bin of 1.1–1.3 nm the ion ratio had a strongest diurnal cycle in summer, when the ratio had a minimum during daytime reflecting the increase in the total particle concentration (see Fig. 6). In the next size bin, 1.3–1.5 nm, the ratio did not have a clear diurnal cycle in any seasons. However, in the two largest size bins, 1.5–2 nm and 2–3 nm, the ion ratio decreased during daytime in spring and winter. In spring this decrease may be related to the formation of particles in NPF events, which are frequent at that time of the year. In the size bin of 2–3 nm the ion ratio also had an evening maximum in autumn, which may be linked to the formation of ion clusters observed frequently in the evening-time in Hyytiälä (Junninen et al., 2008; Buenrostro Mazon et al., 2016).

### 3.4 Effects of environmental conditions on sub-3 nm particle concentrations

#### 3.4.1 Sulfuric acid concentration and condensation sink at different sites

To understand the connection between environmental conditions and sub-3 nm particles, we investigated the relation between the daytime median values of sub-3 nm particle concentration and the medians of sulfuric acid concentration and condensation sink. Figure 10 shows the medians of sub-3 nm particle concentration as a function of sulfuric acid concentration (for most sites estimated from a proxy) for different measurement campaigns. At the sites where the median sub-3 nm particle concentrations were highest, i.e. in Nanjing, Shanghai and San Pietro Capofiume, the median sulfuric acid concentrations were also highest (daytime median values $1.5$–$3.6 \times 10^7$ cm$^{-3}$). Apart from this, no clear relation between the medians of sub-3 nm particle concentration and sulfuric acid concentration can be observed. Thus, although the formation of sub-3 nm particles is likely favored in the conditions with high sulfuric acid concentrations, other factors seem to also affect sub-3 nm particle concentrations. The significance of sulfuric acid for the formation of clusters and small particles has been observed earlier in several studies, including both field measurements and laboratory studies (e.g. Weber et al., 1997; Kulmala et al., 2004; Erupe et al., 2010; Sipilä et al., 2010). On the other hand, recent chamber experiments have shown that particle formation and growth processes are very sensitive to the trace level of base compounds, such as ammonia or amines (Kirkby et al., 2011; Almeida et al., 2013; Lehtipalo et al., 2016), which may explain some of the variability in our data. In addition, the uncertainties of the proxies used for estimating sulfuric acid concentrations may affect the results.



Figure 11 illustrates the relation between the daytime medians of sub-3 nm particle concentration and condensation sink in different measurement campaigns. In Nanjing, Shanghai and San Pietro Capofiume, where the median sub-3 nm particle concentrations (and sulfuric acid concentrations) were highest, high values of condensation sink were also detected (daytime medians 0.01–0.07 s$^{-1}$). On the other hand, in Centreville condensation sink was also high (daytime median 0.01 s$^{-1}$) but sub-3 nm particle concentration was low, possibly due to low sulfuric acid concentration (daytime median $9.6 \times 10^4$ cm$^{-3}$, see Fig 10). Therefore, it seems that the concentration of sub-3 nm particles is determined more by the availability of precursor vapors than by the level of condensation sink. This observation is in agreement with the results of previous studies on sub-3 nm particles (Yu et al., 2014; Rose et al., 2015; Kontkanen et al., 2016).

**3.4.2 Correlation between sub-3 nm particle concentrations and environmental variables**

The correlation between sub-3 nm particle concentration and different variables was also studied separately for each measurement site. The correlation coefficients obtained at different sites are shown in Table 5 together with their confidence intervals, which were calculated by taking into account the effect of autocorrelation (Mudelsee, 2010). In addition, in Table 6 correlation coefficients are presented separately for the 1.1–2 nm and 2–3 nm size ranges for Helsinki and Hyytiälä. Note that for Hyytiälä only data from 2015–2016 were used for calculating the correlation coefficients.

Sulfuric acid concentration had a moderate positive correlation with sub-3 nm particle concentration at all sites (Table 5). The correlation coefficient varied between 0.16 and 0.48 being lowest in Hyytiälä and highest in Nanjing. In Hyytiälä the correlation coefficient depended strongly on the studied size range: the particle concentration in the 1.1–2 nm size range did not correlate with sulfuric acid (R = 0.02) but the particle concentration in the size range of 2–3 nm had a positive correlation (R = 0.38) (Table 6). In Helsinki no similar difference between these two size ranges was observed. A moderate positive correlation between sub-3 nm particle concentration and sulfuric acid concentration at different measurement sites has been observed also in previous studies (Kulmala et al., 2013; Yu et al., 2014; Kontkanen et al., 2016). The correlation indicates that sulfuric acid may be one precursor of sub-3 nm particles but they likely also have other precursors. Furthermore, the fact that in Hyytiälä particle concentrations in the sub-2 nm size range do not correlate with sulfuric acid concentration suggest that at least in Hyytiälä the smallest particles (or clusters) may be predominantly formed from vapors other than sulfuric acid. This is in line with the strong seasonal variation of sub-2 nm particles observed at this site, pointing towards the importance of biogenic sources. On the other hand, the uncertainties of the proxies used for estimating sulfuric acid concentrations may also deteriorate the correlations.

A correlation coefficient between sub-3 nm particle concentration and condensation sink was at some sites negative and at other sites positive (Table 5). The strongest negative correlation was observed in Hyytiälä, Brookhaven, and Centreville (R = -0.20–-0.34), and the strongest positive correlation in Puy de Dôme (R = 0.26). Thus, the relation between condensation sink and sub-3 nm particle concentration seem to vary between different environments. The positive correlation observed in Puy de Dôme is likely due to the simultaneous transport of large particles and precursor vapors to the site. In previous studies at high-altitude sites, condensation sink has been observed to be usually higher on NPF event days than on non-event days for the same reason (Boulon et al., 2010; Manninen et al., 2010; Rose et al., 2015). When investigating the correlation with condensation sink separately for the 1.1–2 nm and 2–3 nm size ranges (Table 6), it can be observed that in Hyytiälä the negative correlation with condensation sink was stronger in the size range of 2–3 nm (R = -0.29) than in the smaller size range (R = -0.12). This suggests that condensation sink may limit the growth of sub-2 nm particles to larger sizes.

Interestingly, at some sites sub-3 nm particle concentration had a positive correlation with ambient temperature (Table 5). The correlation was clear especially in Hyytiälä (R = 0.54), San Pietro Capofiume (R = 0.56) and Shanghai (R = 0.44). The positive correlation with temperature imply that at these sites the formation of sub-3 nm particles may be related to biogenic organic


compounds, as their emissions from vegetation usually depend strongly on temperature (Günther et al., 2012). On the other hand, the positive correlation may also reflect the correlation between sub-3 nm particles and solar radiation (discussed below), as temperature and solar radiation generally correlate with each other. A closer look at the correlations in different size ranges shows that in Hyytiälä the positive correlation existed only in the 1.1–2 nm size range (R = 0.61), whereas the particle

concentration in the 2–3 nm range did not correlate with temperature (R = 0.05) (Table 6). This indicates that in Hyytiälä the smallest, sub-2 nm particles may be formed from organic vapors, which is also consistent with the fact that their concentration does not correlate with sulfuric acid. Recently, Kirkby et al. (2016) showed in their chamber study that aerosol particles can be formed from highly oxidized organic compounds in the absence of sulfuric acid. In addition, the condensation of oxidized organic compounds has been observed to dominate particle formation at a high-altitude Alpine site (Bianchi et al., 2016). On

the other hand, in Hyytiälä sulfuric acid is likely needed for the growth of sub-2 nm particles to larger sizes, as numerous field measurements have proven the importance of sulfuric acid in particle formation in the boreal environment (e.g. Sihto et al., 2006; Nieminen et al., 2009; Petäjä et al., 2009; Kulmala et al., 2013). In Helsinki, sub-3 nm particle concentration did not correlate with air temperature in either of the two size ranges (R= -0.02--0.09), which further strengthens the conclusion that biogenic precursors are likely less important for sub-3 nm particles in this environment.

At most sites there was a negative correlation between sub-3 nm particle concentration and RH (Table 5). This was clear in Hyytiälä (R = -0.48) and San Pietro Capofiume (R = -0.51), which is in agreement with the strong positive correlation between particle concentration and temperature at these sites. A negative correlation existed also in Helsinki, Nanjing and the US sites (R = -0.22--0.48). There were no clear differences in correlation coefficients in the 1.1–2 nm and 2–3 nm size ranges in Hyytiälä and Helsinki (Table 6).

Sub-3 nm particle concentration had a positive correlation with global radiation at all sites. The correlation coefficient ranged from 0.31 obtained in Helsinki to 0.55 in Nanjing (Table 5). In Hyytiälä and Helsinki, these correlation coefficients did not greatly differ between the 1.1–2 nm and 2–3 nm size ranges (Table 6). The positive correlation with radiation suggests the importance of photochemical production of precursor vapors and it is consistent with the observed daytime maxima in sub-3 nm particle concentrations (Fig. 4).

Finally, we also investigated the correlation between sub-3 nm particle concentrations and nitrogen oxides (NO and $NO_x$) in Hyytiälä and Helsinki. In Hyytiälä there was no clear relation between particle concentrations and NO, but a negative correlation with $NO_x$ was observed (R = -0.45 in the 1.1–2.1 nm size range and R = -0.05 in the 2–3 nm size range). The negative correlation is likely related to the fact that $NO_x$ concentration is high when there is little radiation, and thus oxidation by OH and photodissociation processes are slow (Lyuobuotseva et al., 2006). In addition, high $NO_x$ concentrations in Hyytiälä

are often linked to anthropogenic pollution episodes. In contrast, in Helsinki sub-3 nm particle concentration had a positive correlation with NO and $NO_x$. The correlation was stronger in the size range of 1.1–2.0 nm (R = 0.51 for NO and R = 0.39 for $NO_x$) than in the size range of 2–3 nm (R = 0.34 for NO and R = 0.31 for $NO_x$). As nitrogen oxides are tracers for traffic emissions, this suggests that the formation of sub-3 nm particles in Helsinki may be linked to the emissions from engines of cars and buses driving near the measurement site. This conclusion is consistent with the observations made on the diurnal cycle

of particle concentration in Helsinki (see Sect 3.2.2). Likewise, it is probable that sub-3 nm particles are formed, at least partly, due to traffic emissions also at other urban sites.

### 3.5 Connection to NPF events

NPF events are characterized by the appearance of a new mode of small (< 25 nm) particles and their subsequent growth to larger sizes (Dal Maso et al., 2005). The frequency of NPF events observed at different sites is shown in Table 2. The event

frequency was highest in San Pietro Capofiume (86% of days) and in Hyytiälä during spring (40% of days). In Hyytiälä, the





NPF event frequency was lower in other seasons (15–19% in summer and autumn, 0% in winter). In Helsinki the event frequency was highest in spring (13%) but lowest in summer (4%). In Puy de Dôme, Brookhaven, Kent, Shanghai and Nanjing, the event frequency was between 17% and 23%. In Centreville the event frequency was only 9%. When studying particle concentrations in different sub-3 nm size bins, it can be observed that in Hyytiälä the concentration of 2–3 nm particles and its

ratio to the concentration of 1.1–2 nm particles was high in spring when NPF events were frequent (see Sect. 3.1.3). In Helsinki, the link between the concentration of 2–3 nm particles and the event frequency was not as clear. It should be noted that the lower NPF event frequency in Helsinki compared to Hyytiälä is likely due to the fact that in Helsinki only the strongest regional NPF events can be observed, due to pre-existing aerosol particles, and therefore the majority of days are so-called 'undefined' days (Hussein et al., 2005).

All in all, the results indicate that the occurrence of NPF events does not depend solely on the concentration of sub-3 nm particles. This indicates that the formation of sub-3 nm particles and their subsequent growth to larger sizes are two separate processes, as suggested already by Kulmala et al. (2000), and the growth occurs only if conditions are favorable. The favorable conditions may be, for instance, high enough concentrations of condensable precursor vapors (e.g. sulfuric acid and low volatile organic compounds) and low enough concentrations of pre-existing aerosol particles which as a sink for small particles. As

the growth from sub-3 nm sizes to larger particles is generally not observed at night, the photochemical production of condensable vapors is likely needed for the initial growth of particles. In the earlier studies discussing sub-3 nm particle concentrations in Brookhaven and Kent (Yu et al., 2014), Shanghai (Xiao et al., 2015) and Nanjing (Yu et al., 2016), it has been concluded that the increase in sub-3 nm particle concentration observed in daytime does not always lead to an NPF event. For example, Xiao et al. (2016) found that sub-3 nm particles were able to grow to larger sizes only when aerosol surface area

was low and sulfuric acid concentration moderate. In other environments the conditions limiting the growth of particles may be different and determining them is not within the scope of this study. Still, studying the concentration of sub-3 nm particles separately in different size bins seems to be essential to understand the dynamics of sub-3 nm particles and their connection to NPF events.

**4 Conclusions**

In this study, the concentrations of sub-3 nm particles were investigated at nine sites around the world. The particle concentrations were measured with a PSM, together with a DMPS or SMPS. The concentration of sub-3 nm particles was observed to vary significantly at each measurement site and between different environments. The highest sub-3 nm particle concentrations were detected at the sites with the strongest anthropogenic influence, i.e. in Nanjing and Shanghai, China, San Pietro Capofiume, Italy, and Helsinki, Finland. Sub-3 nm particle concentrations were lower at a boreal forest site in Hyytiälä,

Finland, at a high-altitude site in Puy de Dôme, France, and at three sites in the United States, Kent, Brookhaven and Centreville. This indicates that the formation of sub-3 nm particles is favored in the conditions with high concentrations of low volatile precursor vapors originating from anthropogenic pollution sources. When studying the diurnal variation of particle concentrations, sub-3 nm particle concentration was observed to be highest during daytime at all sites. The daytime maxima are likely related to the photochemical production of low volatile precursor vapors and the emissions of precursor vapors, and

possibly also primary particles, from different sources. On the other hand, at most of the sites sub-3 nm particle concentration was relatively high at night, which suggests that sub-3 nm particles can be formed also in the absence of solar radiation.

In Hyytiälä and Helsinki measurements allowed us to study sub-3 nm particle concentration separately in four size bins (1.1–1.3 nm, 1.3–1.5 nm, 1.5–2 nm, and 2–3 nm) in different seasons. In Hyytiälä, sub-3 nm particle concentration exhibited a clear seasonal cycle with the highest concentrations in summer and spring, and the lowest in winter and autumn. In the smallest size

bin the particle concentration was clearly highest in summer, which likely results from strong photochemical reactions and



high concentrations of biogenic organic compounds at this time of the year. In the largest size bin the particle concentration was highest in spring, showing that at that time of the year the conditions in Hyytiälä are most favorable for the growth of particles. In Helsinki, the differences in sub-3 nm particle concentrations between different seasons were less obvious, and high concentrations were observed also in winter in all four size bins. Thus, in Helsinki the formation of sub-3 nm particles is

likely connected to vapors originating from anthropogenic sources, whereas in Hyytiälä biogenic sources are probably more important.

In addition to PSM measurements, at some of the sites the measurements with a NAIS were conducted, which enabled us to study the ratio of ion concentration to the total sub-3 nm particle concentration. The ion ratios were observed to be low at sites where the total sub-3 nm particle concentrations were high. In Hyytiälä the ion ratio was relatively low in summer and spring

but high in winter and autumn. In winter and autumn the ion ratio often exceeded unity, which shows that the PSM was not able to detect all sub-3 nm particles. When studying different sub-3 nm size bins in Hyytiälä, the ion ratio was observed to be highest in the sub-2 nm size bins. All in all, the results imply that neutral particles dominate sub-3 nm particle concentrations in polluted environments and in boreal forest during spring and summer. However, determining the ion ratios more reliably would require more knowledge about the properties of sub-3 nm particles and their activation in the PSM in different

conditions. Also, more simultaneous measurements with the PSM and ion spectrometers should be performed in different environments.

The effect of environmental conditions on sub-3 nm particle concentrations was also investigated. The concentration of sulfuric acid, estimated for most sites from a proxy, was observed to be highest at the sites with high sub-3 nm particle concentration. On the other hand, condensation sink was also highest at these sites, which indicates that the concentration of sub-3 nm particles

is determined by the availability of precursor vapors rather than the value of the sink. When studying correlations between particle concentrations and different variables, sub-3 nm particle concentration was observed to have a positive correlation with sulfuric acid concentration and solar radiation. The correlation with condensation sink was positive at some measurement sites and negative at others. In addition, at some sites sub-3 nm particle concentration showed a positive correlation with temperature. This was clear particularly in Hyytiälä in the sub-2 nm size range, which further suggests the importance of

biogenic sources of precursor vapors in boreal forest. On the other hand, in Helsinki sub-3 nm particle concentration correlated with nitrogen oxides, which indicates that sub-3 nm particles observed at this site may be linked to traffic emissions.

When studying the connection between sub-3 nm particle concentrations and NPF events, it was concluded that the occurrence of NPF events is not determined only by the concentration of sub-3 nm particles. Thus, the formation of particles and their further growth should be considered as two separate processes. Altogether, our results demonstrate that to better understand

the dynamics of sub-3 nm particles, long-term measurements of sub-3 nm particle concentrations, preferably separately in different size bins, are needed. Such measurements should be conducted in different environments and ecosystems, also including the Southern Hemisphere and polar areas. Finally, instrumental development is essential to ensure the reliability of the measured concentrations, especially in the sub-2 nm size range, and to characterize the composition of detected particles.




**Acknowledgements**

This research has received funding from the Academy of Finland Centre of Excellence program (grant nos. 1118615 and 272041), the European Research Council (ERC) project ATM-NUCLE (grant no. 227463), the European Union's Horizon 2020 research and innovation programme projects ACTRIS-2 (grant no. 654109) and nano-CAVa (Marie Sklodowska Curie

grant no. 656994), the European Commission under the Framework Programme 7 project PEGASOS (grant no. 265148). SHL acknowledges funding from National Science Foundation (AGS-1137821; AGS-1241498) and Vijay Kanawade, Janek Uin and You Yi for the help in collecting the PSM data at the US sites. HY acknowledges funding from NSFC 41405116, Jiangsu Province NSF BK20140989 and Jiangsu Specially Appointed Professor grant. WN and AD acknowledge funding from National Natural Science Foundation of China (D0512/41675145) and Jiangsu Collaborative Innovation Center for Climate

Change. LW acknowledges funding from National Natural Science Foundation of China (grant nos. 21222703 and 21561130150) and the Royal Society-Newton Advanced Fellowship (NA140106).

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





**Table 1. Overview of the measurements analyzed in this study.**

| Measurement site | Instruments | Time period | Size range (nm) |
|---|---|---|---|
| Hyytiälä (HTL 10 aut) | PSMproto*, DMPS, NAIS | 4.8–27.8.2010 | 1.3–3.0 |
| Hyytiälä (HTL 11 spr) | PSMA09, DMPS, NAIS | 17.3–1.4.2011 | 1.1–3.0 |
| Hyytiälä (HTL 11 aut) | PSMA09, DMPS, NAIS | 23.8–11.9.2011 | 1.1–3.0 |
| Hyytiälä (HTL 12) | PSMA09, DMPS, NAIS | 19.4–9.5.2012 | 1.3–3.0 |
| Hyytiälä (HTL 13) | PSMA10, DMPS, NAIS | 1.5–23.7.2013 | 1.3–3.0 |
| Hyytiälä (HTL 14) | PSMA11, DMPS, NAIS | 3.4–30.5.2014 | 1.0–3.0 |
| Hyytiälä (HTL 15) | PSMA11, DMPS, NAIS | 8.5.2015–30.4.2016 | 1.1–3.0 |
| San Pietro Capofiume (SPC) | PSMA09, DMPS, NAIS | 16.6–9.7.2012 | 1.5–3.0 |
| Puy de Dôme (PDD) | PSMA09, SMPS, NAIS | 16.1–29.2.2012 | 1.3–2.5 |
| Brookhaven (BRH) | PSMA09*, SMPS | 22.7–14.8.2011 | 1.3–3.0 |
| Kent (KNT) | PSMA09*, SMPS | 15.12.2011–6.1.2012 | 1.3–3.0 |
| Centreville (CTR) | PSMA09, SMPS | 1.6–15.7.2013 | 1.1–2.1 |
| Shanghai (SH) | PSMA11 | 25.11.2013–23.1.2014 | 1.3–3.0 |
| Nanjing (NJ) | PSMA11, NAIS | 1.12.2014–31.1.2015 | 1.1–3.0 |
| Helsinki (HEL) | PSMA11, DMPS | 8.1.2015–31.12.2015 | 1.1–3.0 |

* The PSM was not operated in the scanning mode.





Table 2. Medians of sub-3 nm particle concentration, the ratio of ion concentration to the total sub-3 nm particle concentration, sulfuric acid concentration, condensation sink, and the frequency of new particle formation (NPF) events at different measurement sites. Sulfuric acid concentration is estimated from a proxy for all other campaigns except those that are marked with an asterisk (*). The explanations for abbreviations, the measurement periods and the exact size ranges for particle measurements are shown in Table 1. Data from Hyytiälä (HTL) and Helsinki (HEL) are divided into different seasons: spring (spr), summer (sum), autumn (aut), and winter (wint).

| Measurement site | Sub-3 nm particle concentration (cm$^{-3}$) | Ions to all particles ratio | Sulfuric acid concentration (cm$^{-3}$) | Condensation sink (s$^{-1}$) | NPF event frequency (%) |
|---|---|---|---|---|---|
| HTL spr | 2.9E+03 | 0.16 | 1.0E+06 | 2.6E-03 | 40 |
| HTL sum | 2.0E+03 | 0.33 | 2.4E+05 | 3.6E-03 | 19 |
| HTL aut | 7.9E+02 | 0.83 | 2.6E+05 | 2.0E-03 | 15 |
| HTL wint | 5.8E+02 | 0.71 | 6.9E+05 | 2.1E-03 | 0 |
| SPC | 8.5E+03 | 0.004 | 1.0E+07 | 1.2E-02 | 86 |
| PDD | 5.0E+02 | 0.60 | 3.8E+06 | 3.6E-03 | 23 |
| BRH | 8.0E+02 | - | 3.3E+05* | 6.7E-03 | 17 |
| KNT | 4.7E+02 | - | 9.4E+05* | 6.7E-03 | 22 |
| CTR | 5.9E+02 | 0.47 | 4.0E+04* | 1.5E-02 | 9 |
| SH | 8.5E+03 | - | 3.1E+07 | 7.6E-02 | 21 |
| NJ | 1.7E+04 | 0.02 | 2.0E+07 | 2.7E-02 | 20 |
| HEL spr | 7.8E+03 | - | 2.0E+06 | 4.1E-03 | 13 |
| HEL sum | 5.1E+03 | - | 2.5E+06 | 5.3E-03 | 4 |
| HEL aut | 4.1E+03 | - | 9.2E+05 | 4.3E-03 | 12 |
| HEL wint | 6.9E+03 | - | 2.2E+05 | 3.6E-03 | 8 |

*Sulfuric acid concentration was measured.





**Table 3. Medians of sub-3 nm particle concentration, the ratio of ion concentration to the total sub-3 nm particle concentration, sulfuric acid concentration, condensation sink, and the frequency of new particle formation (NPF) events in Hyytiälä during different measurement campaigns. Sulfuric acid ($H_2SO_4$) concentration is estimated from a proxy for all other campaigns except in spring 2011.**

| Measurement campaign | Sub-3 nm particle concentration (cm$^{-3}$) | Ions to all particles ratio | Sulfuric acid concentration (cm$^{-3}$) | Condensation sink (s$^{-1}$) | NPF event frequency (%) |
|---|---|---|---|---|---|
| HTL 10 aut | 6.7E+02 | 0.56 | 4.1E+05 | 3.1E-03 | 8 |
| HTL11 spr | 2.7E+03 | 0.17 | 1.1E+06* | 1.8E-03 | 75 |
| HTL 11 aut | 7.2E+02 | 0.76 | 6.9E+05 | 2.8E-03 | 10 |
| HTL 12 spr | 3.2E+03 | 0.10 | 4.1E+05 | 2.2E-03 | 62 |
| HTL 13 sum | 4.5E+03 | 0.10 | 2.0E+05 | 3.8E-03 | 37 |
| HTL 14 spr | 5.4E+03 | 0.09 | 3.8E+05 | 2.9E-03 | 48 |
| HTL 15 sum | 2.1E+03 | 0.37 | 2.1E+05 | 3.6E-03 | 12 |
| HTL15 aut | 1.0E+03 | 0.83 | 2.2E+05 | 2.0E-03 | 17 |
| HTL 15 wint | 5.8E+02 | 0.71 | 6.9E+05 | 2.1E-03 | 0 |
| HTL 16 spr | 9.4E+02 | 0.57 | 4.0E+05 | 2.6E-03 | 13 |

*Sulfuric acid concentration was measured.



**Table 4. Medians of the total particle concentration and the ratio of ion concentration to the total particle concentration in four size bins (1.1–1.3 nm, 1.3–1.5 nm, 1.5–2 nm, and 2–3 nm) in Hyytiälä (HTL) in 2011 and 2015–2016, and in Helsinki (HEL) in 2015. Data are divided into different seasons: spring (spr), summer (sum), autumn (aut), and winter (wint).**

| Measurement site | Particle concentration (cm$^{-3}$) | | | | Ions to all particles ratio | | | |
|---|---|---|---|---|---|---|---|---|
| Size range (nm) | *1.1–1.3* | *1.3–1.5* | *1.5–2.0* | *2.0–3.0* | *1.1–1.3* | *1.3–1.5* | *1.5–2.0* | *2.0–3.0* |
| HTL spr | 457 | 160 | 78 | 300 | 0.75 | 1.02 | 0.78 | 0.03 |
| HTL sum | 1129 | 216 | 133 | 194 | 0.34 | 1.25 | 0.89 | 0.06 |
| HTL aut | 384 | 150 | 90 | 109 | 1.05 | 1.45 | 1.02 | 0.07 |
| HTL wint | 235 | 68 | 42 | 172 | 1.05 | 1.36 | 0.79 | 0.03 |
| HEL spr | 2324 | 668 | 747 | 2214 | - | - | - | - |
| HEL sum | 1227 | 386 | 411 | 1870 | - | - | - | - |
| HEL aut | 1134 | 373 | 325 | 1901 | - | - | - | - |
| HEL wint | 2059 | 860 | 721 | 2104 | - | - | - | - |





**Table 5. Pearson's correlation coefficients between sub-3 nm particle concentration and other variables at different measurement sites. The confidence intervals for the coefficients are shown in parenthesis.**

| Measurement site | Sulfuric acid conc. | Condensation sink | Temperature | RH | Radiation |
|---|---|---|---|---|---|
| HTL | 0.16 | -0.22 | 0.54 | -0.48 | 0.43 |
| | (0.10–0.21) | (-0.29--0.16) | (0.49–0.59) | (-0.53--0.42) | (0.38–0.48) |
| SPC | 0.43 | 0.05 | 0.56 | -0.51 | 0.54 |
| | (0.28–0.56) | (-0.08–0.18) | (0.46–0.65) | (-0.60--0.40) | (-0.42--0.63) |
| BRH | 0.44 | -0.34 | 0.29 | -0.41 | - |
| | (0.33–0.53) | (-0.46--0.20) | (0.33–0.53) | (-0.53--0.29) | |
| KNT | 0.37 | 0 | -0.01 | -0.32 | 0.46 |
| | (0.27–0.50) | (-0.15–0.15) | (-0.17–0.15) | (-0.46--0.18) | (0.33–0.58) |
| CTR | 0.31 | -0.20 | 0.24 | -0.22 | - |
| | (0.24–0.39) | (-0.28--0.12) | (0.15–0.32) | (-0.30--0.13) | |
| PDD | 0.37 | 0.26 | 0.12 | 0 | 0.41 |
| | (0.18–0.54) | (0.13–0.38) | (-0.02–0.25) | (-0.13–0.14) | (0.29–0.51) |
| SH | 0.27 | 0.03 | 0.44 | 0 | 0.34 |
| | (0.05–0.47) | (-0.12–0.17) | (0.30–0.56) | (-0.16–0.15) | (0.21–0.46) |
| NJ | 0.48 | -0.16 | 0.22 | -0.48 | 0.55 |
| | (0.34–0.61) | (-0.31–0) | (0.06–0.38) | (-0.60--0.33) | (0.42–0.65) |
| HEL | 0.26 | 0.15 | -0.05 | -0.23 | 0.31 |
| | (0.20–0.31) | (0.10–0.20) | (-0.11–0.01) | (-0.28--0.17) | (0.26–0.36) |





**Table 6. Pearson's correlation coefficients between particle concentration in the size ranges of 1.1–2 nm and of 2–3 nm and other variables in Hyytiälä (HTL) and Helsinki (HEL) in 2015–2016. The confidence intervals for the coefficients are shown in parenthesis.**

| Site and size range | Sulfuric acid conc. | Condensation sink | Temperature | RH | Radiation | NO conc. | NO$_x$ conc. |
|---|---|---|---|---|---|---|---|
| HTL 1.1–2 nm | 0.02 (-0.03–0.08) | -0.12 (-0.18–-0.06) | 0.61 (0.56–0.65) | -0.40 (-0.45–-0.34) | 0.37 (0.32–0.41) | -0.14 (-0.17–-0.11) | -0.45 (-0.49–-0.41) |
| HTL 2–3 nm | 0.38 (0.33–0.42) | -0.29 (-0.33–-0.25) | 0.05 (0.01–0.10) | -0.44 (-0.48–-0.41) | 0.33 (0.29–0.37) | 0.13 (0.10–0.16) | -0.05 (-0.09–-0.01) |
| HEL 1.1–2 nm | 0.24 (0.18–0.30) | 0.10 (0.05–0.15) | -0.09 (-0.15–-0.04) | -0.22 (-0.27–-0.16) | 0.29 (0.25–0.34) | 0.51 (0.47–0.54) | 0.39 (0.35–0.43) |
| HEL 2–3 nm | 0.25 (0.21–0.29) | -0.03 (-0.06–0.01) | -0.02 (-0.06–0.01) | -0.24 (-0.27–-0.21) | 0.26 (0.23–0.29) | 0.34 (0.31–0.37) | 0.31 (0.27–0.34) |





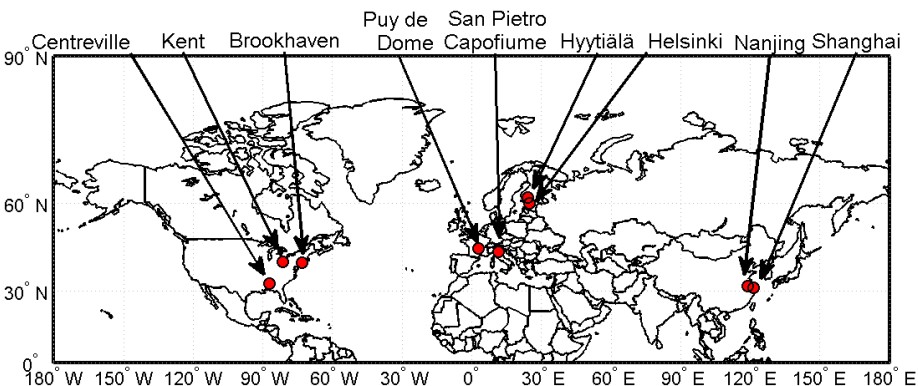

**Figure 1. A map showing the locations of the measurements sites.**





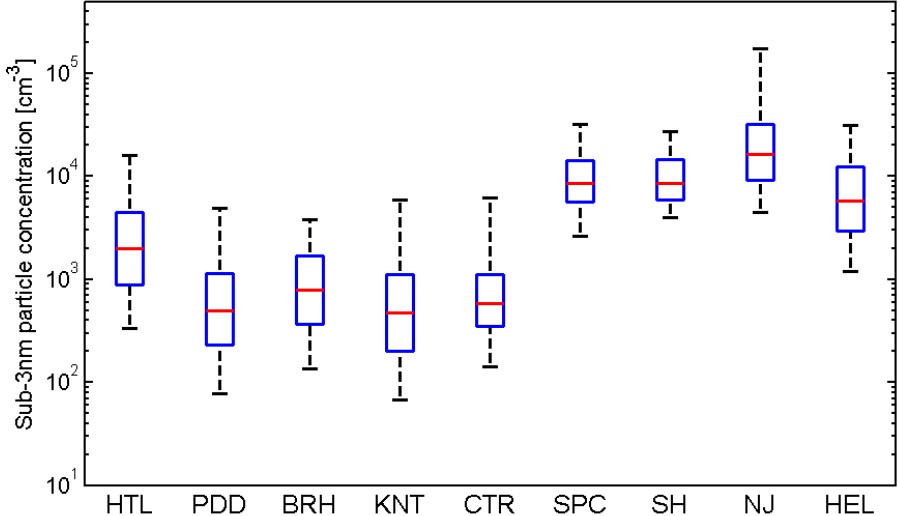

**Figure 2. The variation of sub-3 nm particle concentration at different measurements sites. The red lines show the medians, the blue boxes indicate the 25th and 75th percentiles, and the vertical bars show the 5th and 95th percentiles.**





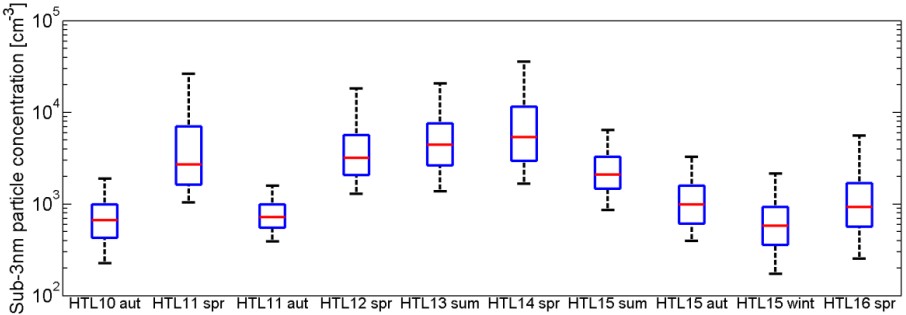

**Figure 3. Sub-3 nm particle concentrations in Hyytiälä during different measurement campaigns. The red lines show the medians, the blue boxes indicate the 25th and 75th percentiles, and the vertical bars show the 5th and 95th percentiles. Note that the data from 2015–2016 are divided into different seasons: summer (sum), autumn (aut), winter (wint), and spring (spr).**





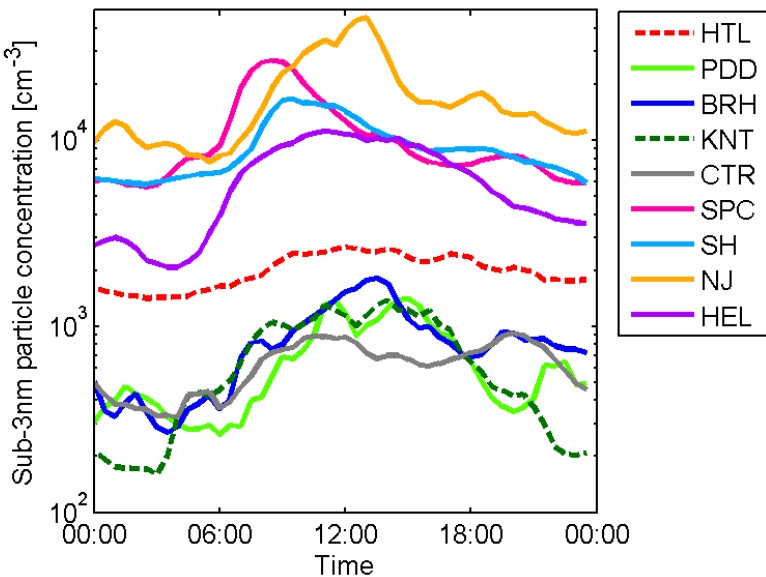

**Figure 4. The median diurnal variation of sub-3 nm particle concentration at different measurement sites.**



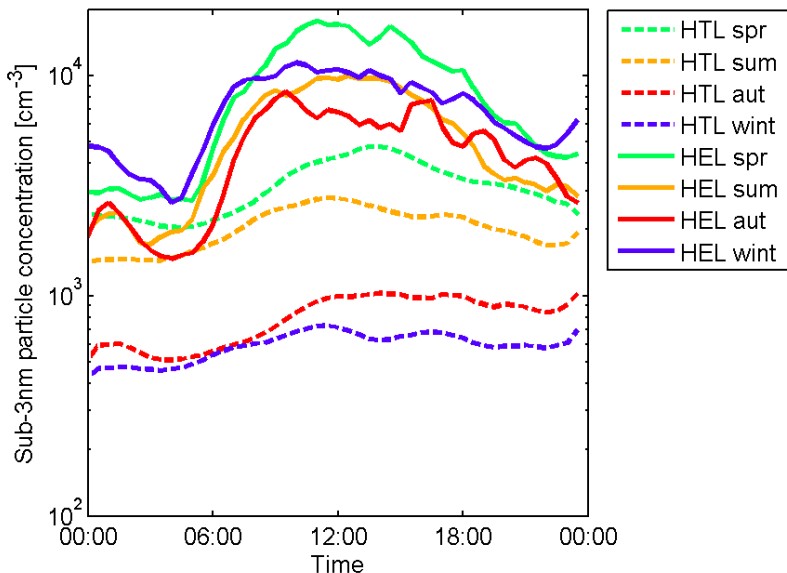

**Figure 5. The median diurnal variation of sub-3 nm particle concentration in Hyytiälä (HTL; dashed lines) and in Helsinki (HEL; solid lines) in different seasons: spring (spr), summer (sum), autumn (aut), and winter (wint).**





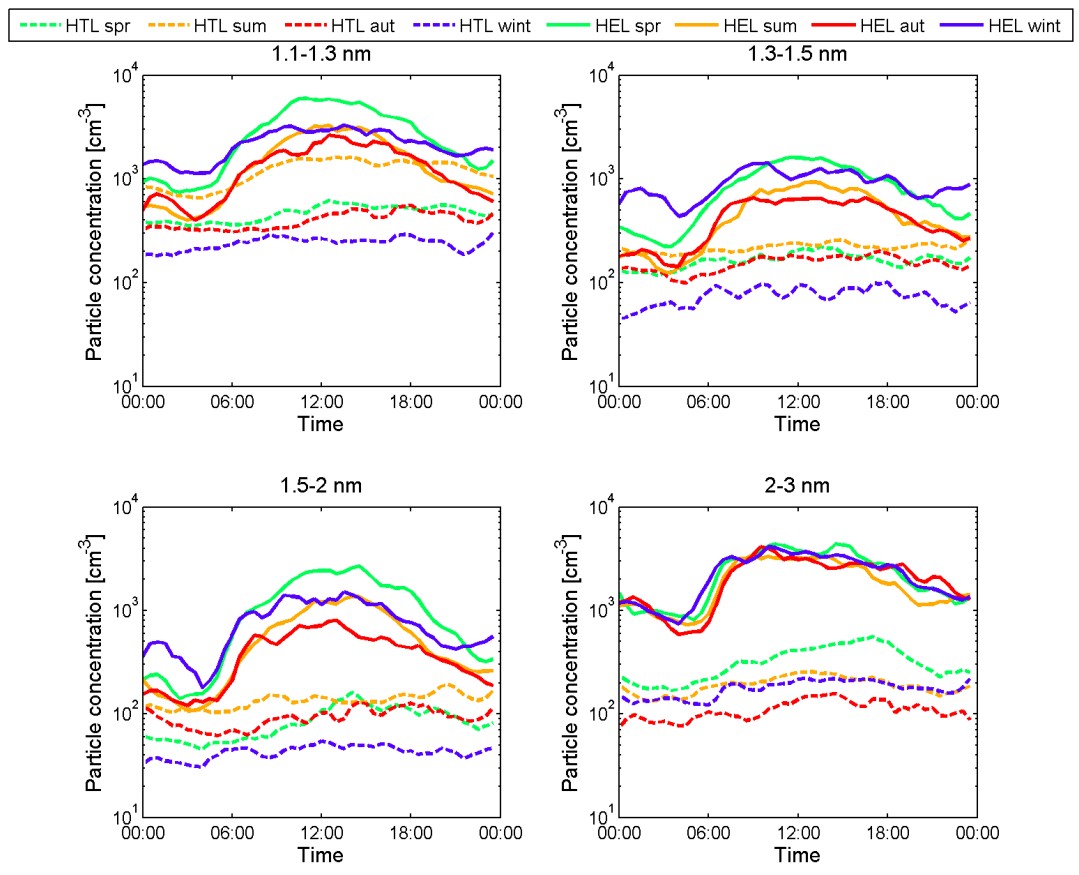

**Figure 6. The median diurnal variation of particle concentration in four size bins (1.1–1.3 nm, 1.3–1.5 nm, 1.5–2 nm, and 2–3 nm) in Hyytiälä (HTL; dashed lines) in 2011 and 2015–2016 and in Helsinki (HEL; solid lines) in 2015. The data are divided into different seasons: spring (spr), summer (sum), autumn (aut), and winter (wint).**





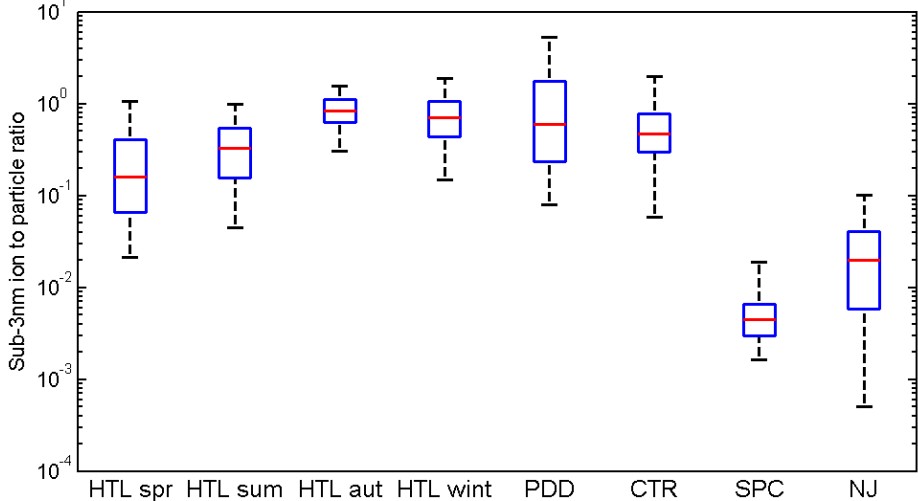

**Figure 7. The ratio of sub-3 nm ion concentration to the total particle concentration at different measurement sites. The red lines show the medians, the blue boxes indicate the 25th and 75th percentiles, and the vertical bars show the 5th and 95th percentiles. The data from Hyytiälä (HTL) are divided into different seasons: spring (spr), summer (sum), autumn (aut), and winter (wint).**





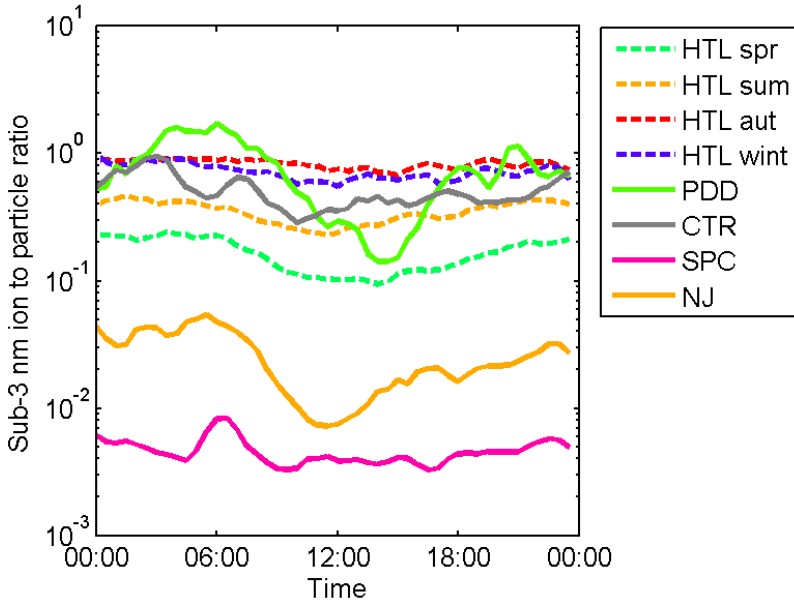

**Figure 8. The median diurnal variation of the ratio of sub-3 nm ion concentration to the total particle concentration during different measurement campaigns. The data from Hyytiälä (HTL) are divided into different seasons: spring (spr), summer (sum), autumn (aut), and winter (wint).**





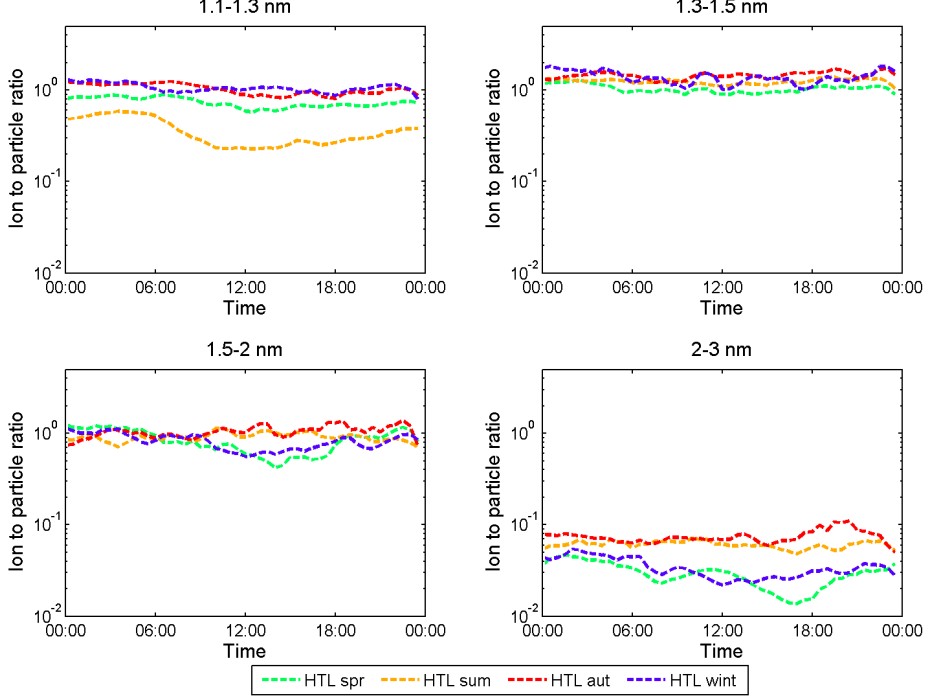

**Figure 9. The median diurnal variation of the ratio of ion concentration to the total particle concentration in four size bins (1.1–1.3 nm, 1.3–1.5 nm, 1.5–2 nm, and 2–3 nm) in Hyytiälä in 2011 and 2015–2016. The data are divided into different seasons: spring (spr), summer (sum), autumn (aut), and winter (wint).**





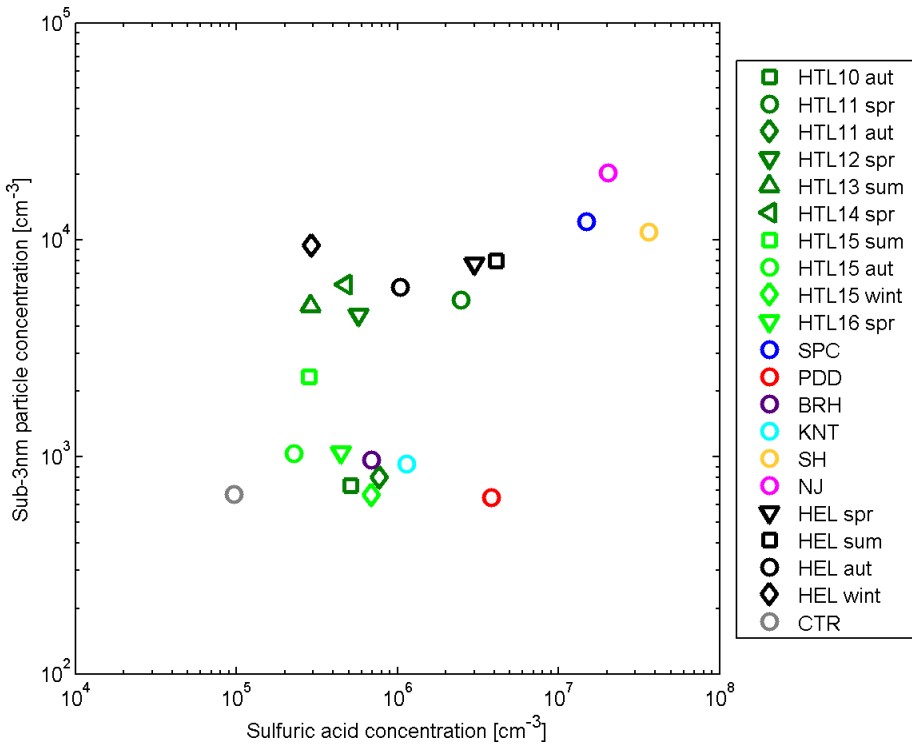

**Figure 10. The daytime medians of sub-3 nm particle concentration as a function of sulfuric acid concentration in different measurement campaigns. Sulfuric acid concentration was calculated from a proxy for all other measurement campaigns except those in Kent, Brookhaven, Centreville and in Hyytiälä during spring 2011.**





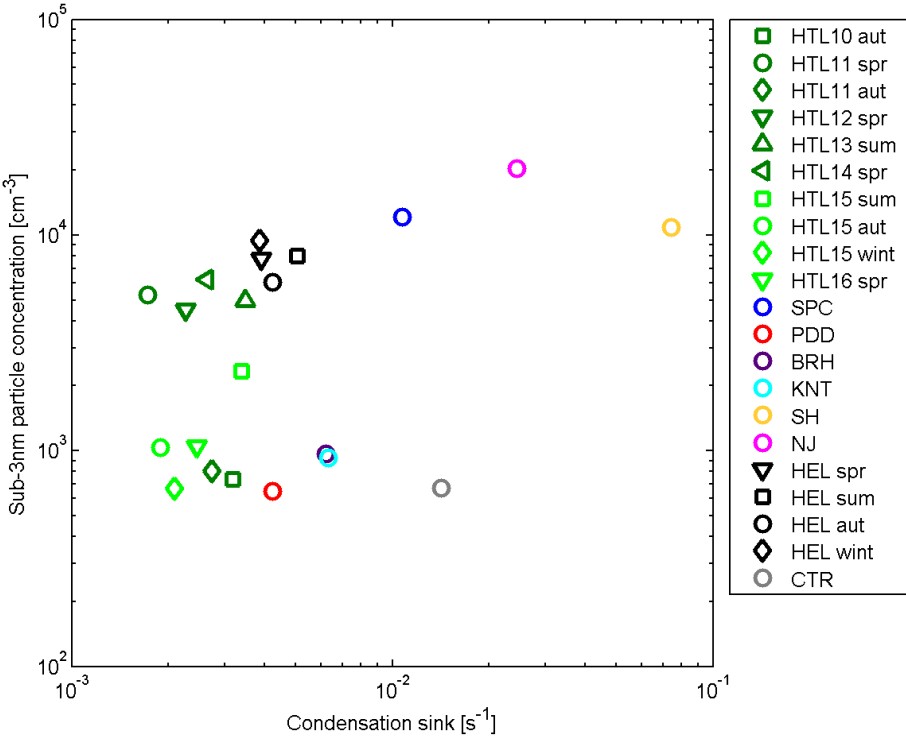

Figure 11. The daytime medians of sub-3 nm particle concentration as a function of condensation sink during different measurement campaigns.