# Peer review of "Measurements of sub-3 nm particles using Particle Size Magnifier in different environments: from clean mountain top to polluted megacities"

_Atmospheric Chemistry and Physics, 2016_

## Referee Comment (RC1) · Anonymous Referee #1 · 7 Nov 2016

The manuscript entitled "A global view on atmospheric concentrations of sub-3 nm particles measured with the Particle Size Magnifier" by Kontkanen et al. summarizes PSM measurements from 9 sites in North America, Europe and Asia. The study compares substantially different ambient settings ranging from highly polluted Asian mega cities to boreal forests and a mountain site. The data suggest a diurnal trend in sub-3 nm particle concentrations in all places and indicate higher sub-3 nm concentrations in polluted areas than in rural background conditions. Therefore the authors conclude that for the existence of sub-3 nm particles the availability of precursor vapors is more important than the level of sink from pre-existing particles. Implications for new particle formation are also discussed. While I think that the topic itself is clearly of sufficient

significance for the scientific community to justify publication in Atmospheric Chemistry and Physics (ACP) I have major concerns about the scientific quality of the manuscript. Below is a list of points which in my opinion should be carefully taken into consideration when revising the manuscript.

As indicated by the title the manuscript aims at a global interpretation of sub-3 nm particle concentrations. One can of course argue that data from only nine measurement sites are no acceptable justification to claim global views. However, what I found even more disappointing is the distribution of sites. All sites are located within a narrow band between 30°N and 60°N (mid-latitudes of the northern hemisphere). The authors themselves have recognized this shortcoming and suggest extension of measurements at the end of the conclusions section. Furthermore, the heterogeneity among these sites (mountain station, forest, cities) suggests to me that an extrapolation to global interpretations is hard to justify. Even if trends are found similar the reasons for this might be quite different. This also reflects in a statement on page 9, lines 29 and 30 where the PSM response in different environments is questioned. Needless to say that data from some stations have been obtained from three weeks measurement campaigns only and in different seasons, respectively. Some of the conclusions are clearly biased by data taken from the long-term measurements in Hyytiälä and Helsinki. From my point of view the data presented here are not sufficient to infer global conclusions.

I would strongly suggest giving the manuscript a different direction and focusing more on data assimilation between different PSMs. Given the effort that was obviously needed to arrive at the numbers presented here that would much better reflect the current work. Especially, since the authors clearly state in the description of the PSMs on page 4, line 8 onwards, that each instrument uses individual settings and needs separate calibration. Interestingly, when problems with unphysical data are identified such as on page 12, line 15 and following, the reason is attributed to the performance of the PSM. I consider this argument fair enough and consistent with my expectations. The intercomparison and reliability of PSM data needs critical treatment. Therefore I

think for this study it would be more appropriate to compare rural background conditions with sites heavily influenced by anthropogenic activities.

Some technical comments: In subsection 2.2.7, site description of Centreville, the authors mention in detail precursors and their emission rates (biogenic and anthropogenic). I assume these numbers have been published somewhere and should therefore be referenced.

Also in subsection 3.4.1, page 13, line 29, a reference regarding the sulfuric acid concentration estimate from a proxy would be desirable. The same applies to line 1 on page 14 where the condensation sink is mentioned. A brief description would be good, how condensation sink was determined, or it should be referenced properly.

Page 15, line 14: I suggest rephrasing this line to "...less important for sub-3 nm particles in urban environment."

Page 16, line 14: "... aerosol particles which act as a sink for small particles."

Figures 4, 5, 6, 8 and 9: If these are median diurnal variations of certain concentrations I would expect that the concentration values should agree at midnight (left and right end of the plots). What does it mean if the concentrations are different? E.g., in Fig. 5, the red dashed line shows a factor ~2 higher concentration than 24 hours before. Shouldn't that be the same?

---

## Referee Comment (RC2) · Anonymous Referee #2 · 14 Nov 2016

This paper compares measurements of sub-3nm particles from nine different locations. Overall the paper is well written the topic is relevant to the ACP audience. The paper discusses the current available data collected with the Particle Size Magnifier (PSM). Some sites have long-term measurements, whereas others are short field campaigns (1-2 months). Several of these studies are currently available in the literature. The value of this paper is comparing data from the different sites (albeit with the constraints discussed below).

I do have several major concerns to be address prior to publication.

1. As stated, the PSM measurements were collected by different research groups using different instruments. To my knowledge, there has not been a comparison study

between the PSM measurements within this paper (or the inlet sampling systems). As stated within the paper, the differences in the lowest size cut-off can affect the comparability of the data.

One of the main conclusions of the paper is that sub 3nm particles are highest at sites with strong anthropogenic influence (Nanjing, Shanghai, and San Pietro Campofiume). Yet, measurements at the North American site on Long Island, which does have strong anthropogenic influence, demonstrated sub 3nm concentration that was significantly less than at the remote Hyytiälä station. Is it possible that these differences are within the uncertainty of comparing the different measurements, and thus have little physical value?

Statements within the paper need to reflect the uncertainty of comparison. For example, I do expect the comparison between the measurements made at Hyytiälä and Helsinki to be valid, where the data inversion and inlet system was identical, including a core sampling probe and automatic background measurements. Yet, what is the specific impact in the comparison at other sites, after considering the differences between the data inversion techniques and inlet system? The ratios between ion spectrometers and PSM measurements greater than 1 further emphasize concerns pertaining to the measurement uncertainty.

A more quantitative description of uncertainty pertaining to the instrument intercomparison is required to provide a global view of the sub-3nm particles. Instead, this paper represents a review of current measurements available, and provides strong justification for intercomparisons of PSMs and development of a global standardized measurements and calibration technique.

Overall, there is value in comparing trends observed between the sites (rather than focusing on absolute values of concentration). I would encourage the authors to rework the paper to reflect.

2. For section 3.4.2 – Correlation between sub-3nm particle concentration and envi-

ronmental variables. Throughout this section, the correlation coefficients (R) are listed, and the confidence interval ranges are found in Table 5. Please add a description of exactly how the confidence interval range was calculated.

Without reasonable confidence in the correlation, there is no reason for a meaningful discussion pertaining to potential physical explanation. For example, the correlation between sub 3nm particles and condensation sink at Puy de Dome had a Pearson's correlation of 0.26 ($R^2$ = .07) with a listed confidence interval of (0.13 − 0.38) was explained with transport. This correlation is meaningless and overemphasized. This section should be greatly reduced and only statistically significant correlations should be acknowledged (i.e. correlations with confidence at the 95% level). In other words, significance level is chosen before data analysis, and typically set to 5%.
* * *

---

## Referee Comment (RC3) · Anonymous Referee #3 · 28 Nov 2016

It is a useful study because there is not enough data in the literature about the number concentrations of particles in the size range below 3 nm. The study puts together data from several locations in Europe, in the USA, and in China. All sites a located on the northern hemisphere, and, therefore, I would avoid using the word "global" in the title of the paper.

Some sites can be characterized as background while others urban, one mountain site is also presented. Not only total number concentrations were studied but also their diurnal patterns, the ratios of charged/uncharged particles, and individual size fractions within the 1-3 nm size range. At some locations longer time series were measured, covering all seasons of the year, at other locations only shorter, several weeks long

campaigns were carried out. A comparability of data between different locations and campaigns is rather limited, because different combinations of instruments were used at different locations, not only different models of particle size magnifiers (prototype, A09, A10, A11), but also different versions of mobility particle sizes (DMPS, SMPS, twin systems). Even within the class of PSM A11, different selection of size classes in the scanning mode was used. On the site with the longest time series (Hyytiala), the different versions of particle size magnifier were used subsequently as they were developed from the prototype until the most recent A11. The authors of the study are aware of the drawbacks mentioned above and tried to compensate for them by selecting the methods of data evaluation and comparison. Therefore I do recommend the paper for publication only with some minor revisions.

On page 3, lines 25-26, the authors state, that the detection limit of particle size magnifier differs for neutral and charged particles by about 0.5 nm in the d50. It would help the reader if the authors add a commentary of how they took this fact into account when they compared the particle number concentrations and concentrations of ions in the size range below 3 nm. In the size range of 1-3 nm, the uncertainty of 0.5 nm covers 25% of this size range.

In the description of individual measurement sites the authors always give a description of particle size magnifier used and usually add a description of the DMPS/SMPS systems. I would recommend that the authors unify these descriptions and add information about sites where this info is missing in the text, for example at PDD, BRH and SH.

At page 8, line 18, the authors say that low concentrations at two locations can be due to technical reasons. Are they aware of these reasons, can they be more specific?

If we take into account that typical uncertainty in the DMPS/SMPS concentration measurements after proper calibration is about 10 %, and this uncertainty can rise substantially going down below 20 nm, the differences in absolute values of number con-

[Figure]

centrations bellow 3 nm are not that significant, keeping in mind that these number concentrations result from subtraction of two larger numbers. At the same time the uncertainty in PSM measurement is affected by its cut diameter d50, related to chemical composition, charging state of particles, and relative humidity. The ratios of charged to neutral particles will be affected little less, while seasonal variations determined at one site using one system, and diurnal variations determined at one site by one system will not be affected by these uncertainties too much.

I would also like to recommend the authors to comment on the fact that there is also a diurnal variation of relative humidity that can also cause diurnal shifts of d50 of particle size magnifiers. Speaking about the concentrations measurements, as a reader I would like to have there some information on calibration of individual systems. I know that the PSM comes with factory calibration but how long after the last calibration was each of the campaigns performed?

---

## Author Comment (AC1) · 11 Jan 2017

**REPLIES TO REFEREES**

We thank the referees for their insightful comments, which have improved our manuscript. We believe that the revised manuscript merits publication in ACP, especially due to the novelty of our data set. This is the first study where observations on sub-3 nm particles are compared at several different measurement sites. Without these measurements, very little would be known about the concentrations of sub-3 nm particles and their variation in different environmental conditions. To address the referees' concerns about reliability of the data, more discussion about the measurement uncertainties was added to the revised manuscript, including error estimations for the cut-off sizes of the PSM. We think that this adds to the value of our study, as in the previous publications on the atmospheric measurements of sub-3 nm particles, measurement uncertainties have not been discussed in such detail.

We have answered to each of the referee's comments below. The bold text is quoted from the referee's comments, and the text in italics has been added to the manuscript. The page and line numbers given in the answers refer to those in the ACPD version of the manuscript.

**Reply to Referee #1**

**The manuscript entitled "A global view on atmospheric concentrations of sub-3 nm particles measured with the Particle Size Magnifier" by Kontkanen et al. summarizes PSM measurements from 9 sites in North America, Europe and Asia. The study compares substantially different ambient settings ranging from highly polluted Asian mega cities to boreal forests and a mountain site. The data suggest a diurnal trend in sub-3 nm particle concentrations in all places and indicate higher sub-3 nm concentrations in polluted areas than in rural background conditions. Therefore the authors conclude that for the existence of sub-3 nm particles the availability of precursor vapors is more important than the level of sink from pre-existing particles. Implications for new particle formation are also discussed. While I think that the topic itself is clearly of sufficient significance for the scientific community to justify publication in Atmospheric Chemistry and Physics (ACP) I have major concerns about the scientific quality of the manuscript. Below is a list of points which in my opinion should be carefully taken into consideration when revising the manuscript.**

**As indicated by the title the manuscript aims at a global interpretation of sub-3 nm particle concentrations. One can of course argue that data from only nine measurement sites are no acceptable justification to claim global views. However, what I found even more disappointing is the distribution of sites. All sites are located within a narrow band between 30_N and 60_N (mid-latitudes of the northern hemisphere). The authors themselves have recognized this shortcoming and suggest extension of measurements at the end of the conclusions section. Furthermore, the heterogeneity among these sites (mountain station, forest, cities) suggests to me that an extrapolation to global interpretations is hard to justify. Even if trends are found similar the reasons for this might be quite different. This also reflects in a statement on page 9, lines 29 and 30 where the PSM response in different environments is questioned. Needless to say that data from some stations have been obtained from three weeks measurement campaigns**

**only and in different seasons, respectively. Some of the conclusions are clearly biased by data taken from the long-term measurements in Hyytiälä and Helsinki. From my point of view the data presented here are not sufficient to infer global conclusions.**

It is true that the measurement sites in this study are all located in the mid-latitudes of the northern hemisphere, and therefore making conclusions on the global scale using this data set is not justified. Therefore, we changed the title of the manuscript to "Measurements of sub-3 nm particles using Particle Size Magnifier in different environments: from clean mountain top to polluted megacities". We also modified the first sentence of the objectives (page 2, line 35) so that the word "global" is not used: *The objective of this study is to provide the first comparison on the concentrations and dynamics of sub-3 nm particles in different environments.*

However, we think that it is an advantage for our study that we have collected data from different environments (e.g. forests, cities and a mountain top) as previously sub-3 nm particle concentrations have been reported only from specific locations and no effort has been made to compare these observations with each other.

**I would strongly suggest giving the manuscript a different direction and focusing more on data assimilation between different PSMs. Given the effort that was obviously needed to arrive at the numbers presented here that would much better reflect the current work. Especially, since the authors clearly state in the description of the PSMs on page 4, line 8 onwards, that each instrument uses individual settings and needs separate calibration. Interestingly, when problems with unphysical data are identified such as on page 12, line 15 and following, the reason is attributed to the performance of the PSM. I consider this argument fair enough and consistent with my expectations. The intercomparison and reliability of PSM data needs critical treatment. Therefore I think for this study it would be more appropriate to compare rural background conditions with sites heavily influenced by anthropogenic activities.**

We agree that measurement uncertainties are not sufficiently discussed in the ACPD version of the manuscript (see also the answers to Referee #2). Therefore, we added a new section (2.2) titled "Measurement uncertainties" to the revised manuscript, where we discuss the different sources of uncertainties in our study: 1) the uncertainties caused by the effects of composition and charging state of particles and environmental conditions on the detection efficiency of the PSM, 2) the uncertainties caused by other instruments used in the study (DMPS, SMPS and NAIS), 3) the uncertainties caused by the differences in measurements between different sites. In addition, we modified the text so that absolute concentration values are not emphasized. We also checked that all the concentration values are presented with the accuracy of only two significant figures. Furthermore, we think that already in the ACPD version of the manuscript we have adapted the clean sites vs polluted sites viewpoint, suggested by the referee. This becomes even clearer in the revised manuscript with the modified title and objectives (see the answer to the previous comment).

**Technical comments**

**In subsection 2.2.7, site description of Centreville, the authors mention in detail precursors and their emission rates (biogenic and anthropogenic). I assume these numbers have been published somewhere and should therefore be referenced.**

A reference was added.

**Also in subsection 3.4.1, page 13, line 29, a reference regarding the sulfuric acid concentration estimate from a proxy would be desirable. The same applies to line 1 on page 14 where the condensation sink is mentioned. A brief description would be good, how condensation sink was determined, or it should be referenced properly.**

The references for sulfuric acid proxy calculations are mentioned in "Supporting data" section (2.2.10 in the ACPD version, 2.3.10 in the revised version). In the same section, it is briefly described how condensation sink is calculated and there is also a reference for that. To remind the reader about this, we now added "*see Sect. 2.3.10*" on page 13, line 29.

**Page 15, line 14: I suggest rephrasing this line to ". . .less important for sub-3 nm particles in urban environment."**

We changed the sentence to *"...less important for sub-3 nm particles in this urban environment"* as in some other urban environments biogenic precursor vapors could be important.

**Page 16, line 14: ". . . aerosol particles which act as a sink for small particles."**

We made this change.

**Figures 4, 5, 6, 8 and 9: If these are median diurnal variations of certain concentrations I would expect that the concentration values should agree at midnight (left and right end of the plots). What does it mean if the concentrations are different? E.g., in Fig. 5, the red dashed line shows a factor 2 higher concentration than 24 hours before. Shouldn't that be the same?**

Yes, it is true that concentration and ion ratio values should agree at midnight in the median diurnal plots. The fact that in the ACPD version of the manuscript some values disagree at midnight is only due to averaging of data before plotting. We now corrected this and changed the median diurnal plots (Figs 4, 5, 6, 8 and 9) to new versions, where the values agree at midnight.

**Reply to Referee #2**

**This paper compares measurements of sub-3nm particles from nine different locations. Overall the paper is well written the topic is relevant to the ACP audience. The paper discusses the current available data collected with the Particle Size Magnifier (PSM). Some sites have long-term measurements, whereas others are short field campaigns (1-2 months). Several of these studies are currently available in the literature. The value of this paper is comparing data from the different sites (albeit with the constraints discussed below). I do have several major concerns to be address prior to publication.**

**1. As stated, the PSM measurements were collected by different research groups using different instruments. To my knowledge, there has not been a comparison study between the PSM measurements within this paper (or the inlet sampling systems). As stated within the paper, the differences in the lowest size cut-off can affect the comparability of the data. One of the main conclusions of the paper is that sub 3nm particles are highest at sites with strong anthropogenic influence (Nanjing, Shanghai, and San Pietro Campofiume). Yet, measurements at the North American site on Long Island, which does have strong anthropogenic influence, demonstrated**

**sub 3nm concentration that was significantly less than at the remote Hyytiälä station. Is it possible that these differences are within the uncertainty of comparing the different measurements, and thus have little physical value?**

**Statements within the paper need to reflect the uncertainty of comparison. For example, I do expect the comparison between the measurements made at Hyytiälä and Helsinki to be valid, where the data inversion and inlet system was identical, including a core sampling probe and automatic background measurements. Yet, what is the specific impact in the comparison at other sites, after considering the differences between the data inversion techniques and inlet system? The ratios between ion spectrometers and PSM measurements greater than 1 further emphasize concerns pertaining to the measurement uncertainty. A more quantitative description of uncertainty pertaining to the instrument intercomparison is required to provide a global view of the sub-3nm particles. Instead, this paper represents a review of current measurements available, and provides strong justification for intercomparisons of PSMs and development of a global standardized measurements and calibration technique. Overall, there is value in comparing trends observed between the sites (rather than focusing on absolute values of concentration). I would encourage the authors to rework the paper to reflect.**

We agree that measurement uncertainties are not emphasized enough in the ACPD version of the manuscript (see also the answers to Referee #1). Therefore, a new section (2.2) was added to the revised manuscript, where we discuss the different sources of uncertainty in our study. In addition, we modified the text so that the focus is not so much on the absolute concentration values. As the referee points out, despite the significant uncertainties, our data set can be used to make general conclusions on the variation of sub-3 nm particles in different environments. Therefore, we are still confident about the main conclusions of our study: 1) the concentrations of sub-3 nm particles are generally higher in polluted than in clean environments, 2) the concentrations are higher during daytime than at night, 3) in boreal forest sub-3 nm particle concentration is higher in summer than in winter, 4) the fraction of ions of all sub-3 nm particles is low in environments with high concentrations of sub-3 nm particles. These general conclusions do not change, even if at some of the measurement sites (e.g. the site on Long Island mentioned by the referee) concentrations were underestimated to some extent due to, for instance, a technical reason. On the other hand, it is true that this data set is too limited to make conclusions on the concentrations of sub-3 nm particles on the global scale, and therefore we do not anymore use the word "global" in the title and text of the revised manuscript. Finally, we agree with the referee that there is clearly a need for standardized procedures for the calibration and measurements performed using the PSM. To emphasize this, we added a following sentence to the end of the conclusions section: *In addition, to enable more accurate comparisons between different measurement sites, standardized procedures for the calibration and measurements of sub-3 nm particles should be established.*

**2. For section 3.4.2 – Correlation between sub-3nm particle concentration and environmental variables. Throughout this section, the correlation coefficients (R) are listed, and the confidence interval ranges are found in Table 5. Please add a description of exactly how the confidence interval range was calculated. Without reasonable confidence in the correlation, there is no reason for a meaningful discussion pertaining to potential physical explanation. For example, the correlation between sub 3nm particles and condensation sink at Puy de Dome had a Pearson's correlation of 0.26 ($R^2$ = .07) with a listed confidence interval of (0.13 – 0.38) was**

**explained with transport. This correlation is meaningless and overemphasized. This section should be greatly reduced and only statistically significant correlations should be acknowledged (i.e. correlations with confidence at the 95% level). In other words, significance level is chosen before data analysis, and typically set to 5%.**

The confidence intervals were calculated using Fisher's r-to-z transformation, which takes into account the fact that the confidence intervals around Pearson's correlation coefficient are not symmetrical. In addition, autocorrelation, which reduces the effective data size (i.e. the number of statistically independent data), was also considered according to Mudelsee (2010). The confidence intervals were calculated at the 95% confidence level. This means that if the confidence interval does not contain 0, the correlation is significant at the 95% confidence level. Therefore, the correlation between sub-3 nm particle concentration and condensation sink in Puy de Dôme, mentioned by the referee, is indeed statistically significant, although the correlation is not very strong. However, we agree that the calculation of the confidence intervals and their meaning are not sufficiently explained in the manuscript. Therefore, we modified the beginning of Section 3.4.2. It now reads (page 14, line 11):

*The correlation coefficients obtained at different sites are shown in Table 5 together with their confidence intervals at the 95% confidence level. The confidence intervals were calculated using Fisher's transformation. The autocorrelation, which reduces the effective data size, was also taken into account (Mudelsee, 2010).*

In addition, we added the confidence level of the confidence intervals in the captions of Table 5 and 6.

**Reply to Referee #3**

**It is a useful study because there is not enough data in the literature about the number concentrations of particles in the size range below 3 nm. The study puts together data from several locations in Europe, in the USA, and in China. All sites a located on the northern hemisphere, and, therefore, I would avoid using the word "global" in the title of the paper. Some sites can be characterized as background while others urban, one mountain site is also presented. Not only total number concentrations were studied but also their diurnal patterns, the ratios of charged/uncharged particles, and individual size fractions within the 1-3 nm size range. At some locations longer time series were measured, covering all seasons of the year, at other locations only shorter, several weeks long campaigns were carried out. A comparability of data between different locations and campaigns is rather limited, because different combinations of instruments were used at different locations, not only different models of particle size magnifiers (prototype, A09, A10, A11), but also different versions of mobility particle sizes (DMPS, SMPS, twin systems). Even within the class of PSM A11, different selection of size classes in the scanning mode was used. On the site with the longest time series (Hyytiala), the different versions of particle size magnifier were used subsequently as they were developed from the prototype until the most recent A11. The authors of the study are aware of the drawbacks mentioned above and tried to compensate for them by selecting the methods of data evaluation and comparison. Therefore I do recommend the paper for publication only with some minor revisions.**

As explained in the answers to other referees, the word "global" is not anymore used in the revised version of the manuscript.

**On page 3, lines 25-26, the authors state, that the detection limit of particle size magnifier differs for neutral and charged particles by about 0.5 nm in the d50. It would help the reader if the authors add a commentary of how they took this fact into account when they compared the particle number concentrations and concentrations of ions in the size range below 3 nm. In the size range of 1-3 nm, the uncertainty of 0.5 nm covers 25% of this size range.**

As discussed in the new "Measurement uncertainties" section (2.2) of the revised manuscript, the cut-off size of the PSM is indeed about 0.2–0.5 nm higher for neutral particles than for charged particles. Unfortunately, it is not possible to take this into account when calculating the ion ratio by comparing the concentrations measured with the PSM to the ion concentrations measured with the NAIS. This is because the sub-3 nm particle population detected with the PSM includes both neutral and charged particles. To remind the reader about the effect of charge on the detection efficiency of the PSM, the following sentences were added to the section where ion ratios are discussed (page 12, line 17): *Especially, it should be noted that charged particles have been observed to be activated in the PSM more efficiently than neutral particles. Therefore, it is possible that the PSM detects certain sized charged particles but not the neutral species of the same physical size.*

**In the description of individual measurement sites the authors always give a description of particle size magnifier used and usually add a description of the DMPS/SMPS systems. I would recommend that the authors unify these descriptions and add information about sites where this info is missing in the text, for example at PDD, BRH and SH.**

We followed the referee's suggestion and unified the descriptions of the DMPS/SMPS systems and added the missing information.

**At page 8, line 18, the authors say that low concentrations at two locations can be due to technical reasons. Are they aware of these reasons, can they be more specific?**

The unexpectedly low concentrations at these sites may be caused by 1) the composition/charge of particles or environmental conditions (especially air humidity), which affect the cut-off size of the PSM, 2) technical reasons such as settings of the PSM or losses in the sampling lines. To clarify this, we added the following sentence, including a reference to the new "Measurement uncertainties" section, to the revised manuscript (page 8, line 18):

*The low concentrations may be due to, for example, the properties of particles or ambient conditions, which can affect the detection efficiency of the PSM, or technical reasons, such as the settings of the PSM or losses in the sampling lines (see also Sect. 2.2).*

**If we take into account that typical uncertainty in the DMPS/SMPS concentration measurements after proper calibration is about 10%, and this uncertainty can rise substantially going down below 20 nm, the differences in absolute values of number concentrations below 3 nm are not that significant, keeping in mind that these number concentrations result from subtraction of two larger numbers. At the same time the uncertainty in PSM measurement is affected by its cut diameter d50, related to chemical composition, charging state of particles, and relative humidity. The ratios of charged to neutral particles will be affected little less, while seasonal variations determined at one site using one system, and diurnal variations determined at one site by one system will not be affected by these uncertainties too much.**

It is true that in addition to the uncertainties of the PSM, the uncertainties of other instruments utilized in the study should be considered. In the revised manuscript, all the sources of uncertainty, including those mentioned by the referee, are discussed in Sect. 2.2. However, as pointed out by the referee, although there are significant uncertainties in the absolute values of sub-3 nm particle concentrations, the diurnal and seasonal cycles of concentrations can be considered more reliable. Therefore, in the revised manuscript, we do not emphasize the concentration values as much as in the previous version, but we focus more on the variation of concentrations and ion ratios on diurnal and seasonal scale (see also the answers to Referees #1 and #2).

**I would also like to recommend the authors to comment on the fact that there is also a diurnal variation of relative humidity that can also cause diurnal shifts of d50 of particle size magnifiers.** We thank the referee for pointing this out. According to our laboratory experiments, the key quantity affecting the detection efficiency of the PSM is the absolute water content of air, not the relative humidity. Therefore, to estimate the effect of varying air humidity on the cut-off size of the PSM, we need to study the diurnal cycle of air dew point temperature. For example, in Hyytiälä, the diurnal variation of dew point temperature is typically less than 10 ℃ (calculated based on the results of Lyubovtseva et al. (2005)). When comparing this to the results of laboratory experiments by Kangasluoma et al. (2013), we can see that the resulting diurnal variation in the cut-off size of the PSM is less than ±0.1 nm. When considering all other sources of uncertainty, the effect of diurnal cycle of air humidity on the cut-off size can thus be considered negligible. However, it should be noted that the seasonal variation of air humidity likely has a larger effect on the cut-off size of the PSM. This was taken into account in long-term measurements conducted in Hyytiälä and Helsinki by performing background measurements and adjusting the settings of the PSM accordingly (see Sect. 2.1). Discussion about the effect of air humidity on the cut-off size of the PSM was added to Sect. 2.2 of the revised manuscript.

**Speaking about the concentrations measurements, as a reader I would like to have there some information on calibration of individual systems. I know that the PSM comes with factory calibration but how long after the last calibration was each of the campaigns performed?** The instruments used in Hyytiälä, Helsinki and San Pietro Capofiume were calibrated right before/after the field measurements. During the long-term measurements in Hyytiälä and Helsinki, the background was regularly measured to verify the calibration, and the settings of instruments were adjusted when needed. For other instruments, calibrations provided by the manufacturer were used.

**References**

Lyubovtseva, Y. S., Sogacheva, L., Dal Maso, M., Bonn, B., Keronen, P., and Kulmala, M.: Seasonal variations of trace gases, meteorological parameters, and formation of aerosols in boreal forest, Bor. Environ. Res., 10, 493–510, 2005.